# Biodegradable Iron and Porous Iron: Mechanical Properties, Degradation Behaviour, Manufacturing Routes and Biomedical Applications

**DOI:** 10.3390/jfb13020072

**Published:** 2022-06-01

**Authors:** Mariana Salama, Maria Fátima Vaz, Rogério Colaço, Catarina Santos, Maria Carmezim

**Affiliations:** 1IDMEC, Instituto Superior Técnico, Departamento de Engenharia Mecânica, Universidade de Lisboa, Av. Rovisco Pais, 1049-001 Lisboa, Portugal; fatima.vaz@tecnico.ulisboa.pt (M.F.V.); rogerio.colaco@tecnico.ulisboa.pt (R.C.); 2ESTSetúbal, CDP2T, Instituto Politécnico de Setúbal, Campos IPS, 2910-761 Setúbal, Portugal; catarina.santos@estsetubal.ips.pt; 3Centro Química Estrutural, IST, Universidade de Lisboa, Av. Rovisco Pais, 1049-001 Lisboa, Portugal

**Keywords:** biodegradable metals, iron, porous iron, additive manufacturing, porous scaffolds

## Abstract

Biodegradable metals have been extensively studied due to their potential use as temporary biomedical devices, on non-load bearing applications. These types of implants are requested to function for the healing period, and should degrade after the tissue heals. A balance between mechanical properties requested at the initial stage of implantation and the degradation rate is required. The use of temporary biodegradable implants avoids a second surgery for the removal of the device, which brings high benefits to the patients and avoids high societal costs. Among the biodegradable metals, iron as a biodegradable metal has increased attention over the last few years, especially with the incorporation of additive manufacturing processes to obtain tailored geometries of porous structures, which give rise to higher corrosion rates. Withal by mimic natural bone hierarchical porosity, the mechanical properties of obtained structures tend to equalize that of human bone. This review article presents some of the most important works in the field of iron and porous iron. Fabrication techniques for porous iron are tackled, including conventional and new methods highlighting the unparalleled opportunities given by additive manufacturing. A comparison among the several methods is taken. The effects of the design and the alloying elements on the mechanical properties are also revised. Iron alloys with antibacterial properties are analyzed, as well as the biodegradation behavior and biocompatibility of iron. Although is necessary for further in vivo research, iron is presenting satisfactory results for upcoming biomedical applications, as orthopaedic temporary scaffolds and coronary stents.

## 1. Introduction

Metallic biomaterials have been used in medical applications for a long time [1], due to unique mechanical properties, wear resistance and easy production. They are critical for many load-bearing functions, however, long-term presence of these metals in the body is associated with an increased risk of development of cutaneous and systemic hypersensitivity reactions. A relatively high modulus of these metals compared to natural bone tissue leads to stress shielding and consequent osteopenia [1]. To overcome these problems, a new generation of metals with in vivo degradation, have been developed and tailored with an appropriated host response and complete metal dissolution known as biodegradable metals (BM). The most researched BM are magnesium (Mg), zinc (Zn) and iron (Fe), due to their good in vivo biocompatibility, controlled degradation profile and sufficient mechanical strength to support and promote bone healing during bone regeneration process [1].

Biodegradable porous metallic biomaterials with adjusted biodegradation behaviour and mechanical properties can be the new ideal bone substitutes in non-load bearing applications. There are possibilities for improvement regarding the production, design, performance and the biodegradation behaviour of those promising metallic biomaterials. Especially for porous iron, that easily can overcome magnesium and zinc due to its tailored surface and composition. Comparisons on the properties of the three biodegradable metals, Mg, Zn and Fe can be found in the literature. For example, Dong et al. [2] present a comparison of the corrosion behaviour of Mg, Zn and Fe. It was concluded that although Mg has suitable mechanical properties in order to avoid stress shielding, it degrades rapidly leading to non-uniform corrosion and to accumulation of H_2_ gas. The corrosion rate of Zn is adequate, between the rate obtained for Mg and Fe, but Zn has no-suitable mechanical properties [2,3]. Iron presents a very slow degradation rate, higher stiffness than bone but possess good biocompatibility [3]. Iron degradation occurs by a corrosion mechanism, due to electrochemical dissolution, which occurs when a metallic sample is in contact with the human body fluids. Several solutions to overcome some of the disadvantages of iron will be further reviewed.

A true reflection of the iron importance for biomedical applications is the large number of recent publications available in the Science Direct database (Figure 1). Considering the keywords “biodegradable metals”, the database has been returned 7950 manuscripts were already published for 2022. Conversely, 13,821 of them published in the year 2021 and 10,499 in 2020. From these, 1614 manuscripts in 2022 were focused on “biodegradable porous iron”, whilst 2584 manuscripts focused on the subject in 2021 and 1639 manuscripts in 2020. The consistent growth of publications (Figure 1) in the last 5 years and the enlarging topic attention on biodegradable iron and porous iron is a reality. These consistent publications increment disclose the great importance and actuality of this subject. It directs attention to the challenge of designing functional and complex architecture like the bone structure, providing essential cellular microenvironment. Indeed, societal and medical demands of temporary medical implants urge research focus on iron, with novel strategies and precise requirements for biomedical applications. To address these critical issues different approaches have been considered. Here, it will be highlighting some of the recent advances in the production of biodegradable porous iron biomaterials for regenerative medicine. Along with the biodegradation profile and mechanisms of porous iron biomaterial with several compositions.

The present paper presented a review on several studies, in the last decade, about biodegradable iron, with the advancements in metallurgy. A particular focus was given to temporary applications as vascular stents and temporary bone fixation devices pointing out emergent trends for near future.

## 2. Fabrication Techniques for Iron and Porous Iron

Although cellular structures have been widely studied driven by the pioneer work of Gibson and Ashby [4], some challenges still remain as the manufacturing of complex geometrical structures. The fabrication method is of utmost importance as it affects the microstructure and consequently the mechanical properties of a certain structure. The purpose of the methods described in the present work is to obtain a porous structure with high surface area. A large variety of manufacturing methods has been reported in the literature to obtain iron porous structures. Some methods will be denoted by “conventional” and other will be designated as “advanced” methodologies as they rely on additive manufacturing procedures.

### 2.1. Conventional Manufacture Techniques

Porous structures of biodegradable metals have been manufactured by traditional methods, such as direct foaming, spray foaming, chemical vapour deposition and electrophoretic deposition, powder metallurgy and melt injection moulding [5,6]. Even though these techniques control some specifications of the pores, they are not accurate in pore dimensions and, as a result, randomly organized porous structures are achievable [7].

The foaming procedures are difficult to apply on iron, due to the high density, high melting point, high surface tension and low viscosity of iron melt [7].

Powder metallurgy is another technique adequate for making porous metallic biomaterials, having the advantage of the fabrication of final-shaped products with an interconnected porous structure, which is useful for bone regeneration applications [7]. Also powder metallurgical techniques are less expensive than 3D-printing or laser sintering [7]. One of the powder metallurgy methods is hot isostatic pressure, which consists of the compaction of metallic powders through an application of a pressure, at a given temperature that compresses and sinters the parts at the same time. The compacting pressure and initial powder size are important parameters that will influence the final structure properties, being the use of finer powders useful to obtain good mechanical properties [7].

The fabrication of biodegradable iron alloy stents with powder methods was first performed by Hermawan et al. [3,8], which developed alloys that degrade faster than pure iron. For example, a Fe-Mn alloy produced by powder methods exhibits a faster in vitro degradation than the same alloy obtained by casting, due to the unavoidable presence of porosity [9]. The addition of silver to Fe-Mn alloys was evaluated with the development of Fe-(30 wt%)Mn-(1–3 wt%)Ag alloys, which were obtained by powder mixtures, mechanically alloying and sintering [10]. The same procedure of powder metallurgy with ball milling of the powders, mechanical allowing and sintering was used by Mandal et al. to develop a novel Fe-Mn-Cu alloy with enhanced antimicrobial properties [11]. The fabrication of iron composites (Fe/Mg_2_Si) was also performed by powder metallurgy [9], as well as the production of Fe–Ag and Fe–Au composites [12]. Gorejová et al. [13] were also able to produce porous structures from the carbonyl iron powder via the powder metallurgy process, in which polyurethane foams were impregnated by the slurry with the iron powder and thermally treated, sintered, to obtain the final structure [13].

Although traditional processing technologies possess advantages, they show poor ability in fabricating parts with complex geometrical shapes, which may be needed for bone implant in order to respond to the patient requirements.

### 2.2. Advanced Manufacture Techniques/Additive Manufacturing

The emergence of additive manufacturing (AM) procedures allowed obtaining parts with a porous structure with a certain shape and geometry, that were difficult to produce through more conventional procedures. Additive manufacturing (AM) has led to a revolutionary change in manufacturing engineering for clinical applications and design for metallic implants [6,14,15,16]. AM is capable of controlling the pore size, interconnectivity, shape, and geometry of the biodegradable metallic scaffold [14]. These advanced techniques are ideal methods for medical applications in comparison with classic methods, being precisely controlled, customized to the patient needs and due to its ability to produce replicas of the CT-imaged tissue [17,18]. The application of AM to nondegradable metals, such as titanium and CoCr alloys has proven to be extremely successful [14]. Still, the use of AM to fabricate biodegradable metallic parts is beginning to be studied [14].

AM techniques produce 3D complex parts in a layer-by-layer sequence from a computer-aided design (CAD) model. For bone substitute devices, porous structures can be designed with tailoring architecture in order to minimize stress shielding, to simplify fluids transport and to promote fast healing [19]. The advantage of tailored porous scaffolds is to provide a structure that mimics the bone, allowing the permeability of physiological fluids and cells ingrowth [14,20]. Additionally, as the porous structures produced by AM have a larger surface area, they generally result in higher degradation rates [21].

Among the metal-based additive manufacturing (MAM) techniques, powder bed fusion (PBF) is the most widely used method for the production of metal implants, along with selective laser melting (SLM) and electron beam melting (EBM) [6,14]. MAM processes behaviour is dependent of transient heat transfer, powder thermal properties and melt pool temperature [6]. While in EBM the energy source is an electron beam, in SLM a laser beam with adjustable wavelength is used [6]. As a consequence, EBM can only be used in conductive metals, while SLM can be used to produce metals, ceramics and polymers [6]. Also, compared to SLM, EBM process has a larger zone affected by the heat, producing larger feature sizes [6].

Selective laser melting and electron beam melting are fast and not so expensive tools to prepare orthopaedic devices, presenting low material waste and feasibility to mix different materials with functional gradient [22]. Both electron beam melting (EBM) and selective laser melting (SLM) are able to fabricate structures with complex architecture [6].

In the literature, other studies have reported the production of iron porous structures by several advanced techniques, such as inkjet 3D printing [23,24]. 3D printing and pressure less microwave sintering [25], direct metal printing [21], laser metal deposition and selective laser melting [26,27,28]. 

Among the solid free-form fabrication methods, inkjet 3D printing has the advantage of manufacture metals, polymers, ceramics and composites [24]. Inkjet 3D process deposits liquid binder selectively onto layers of spread powder creating layers of the parts defined by CAD model [24]. For example, Chou et al. [24] used binder jet printing of Fe-30Mn powders to obtain porous structures that presented excellent cytocompatibility and mechanical properties close to the ones of human bone, reducing the stress-shield effects. These structures may be used in low-load-bearing applications. The 3D printed Fe-30Mn structures were found to corrode faster than pure iron [24].

The fabrication procedure consisting of 3D printing and pressure-less microwave sintering consists in the printing of a polymeric structure on which the mixture of powder is poured. Then, the set is placed in a furnace and with heating and vapourization the polymer vanishes, remaining a porous structure which afterwards, is submitted to microwave sintering process [25]. This method allowed obtaining iron structures with porosities in the range of 45.6–86.9% with ultimate compressive strength of 13.16–52.06 MPa.

Li et al. [21] managed to get successful porous iron structures through direct metal printing of iron powders. The biodegradation behaviour showed that the mechanical properties of the porous structures were E = 1600–1800 MPa after 28 days of biodegradation, close to the values of trabecular bone. Electrochemical results revealed that the rate of biodegradation was 12 times higher for AM porous iron in comparison of that of cold-rolled iron [21].

Carluccio et al. [26] made a comparative study with selective laser melting, laser metal deposition and the traditional technique of casting to manufacture pure iron as biodegradable metal. The authors found that selective laser metal produces hierarchical porosity in complex configurations, being an advantage compared to laser metal deposition [26]. Table 1 provides the mechanical properties for compression tests of samples manufactured with the three mentioned methods. Although the Young’s modulus remains the same for the three methods, the compressive strength is higher for SLM [26].

The microstructural difference between these three types of manufacturing iron structures is mainly the grain size and morphology [26]. The grain size is dependent of the cooling rate, and for casting process, the cooling rate is the slowest, between 10^1^ and 10^2^ K/s [26]. For laser-based additive manufacturing, the cooling rates are usually higher due to small melt pools, creating a finer grain size [26]. The grain size can diverge from SLM and LMD, as presented in Figure 2. The corrosion rate of SLM iron improved significantly and it may be proportional to the smaller average grain size [26].

Due to the significantly higher cooling rates of the laser based AM processes, the grain size is lower which promotes higher mechanical properties of the SLM samples [26]. SLM process for iron presented improved mechanical properties and, based on Li et al. [29] research, iron prostheses manufactured by SLM with hierarchical porosity resulted in Young’s modulus below 20 GPa [26,29].

## 3. Mechanical Properties of Biodegradable Porous Iron

Iron-based biodegradable structures are considered to be alternatives to permanent metallic implants [24,30,31], in non-load bearing applications, as they may exhibit mechanical properties closer to the human bone [32]. Cortical bone, which is called compact bone, is the outside layer of the structure and has a porosity of only 3 to 5%. Meanwhile, cancellous or trabecular bone, is a porous network, with spongy appearance, with porosity between 50 to 90%, and is filled with blood vessels and marrow [6].

Compact iron exhibits a Young’s modulus of 210 GPa [33]. However, cancellous bone Young’s modulus is the range 10–20 GPa, and the trabecular bone Young’s modulus is the interval of 3 × 10^−4^–3 × 10^−3^ GPa [6,34]. The use of compact iron in implants may cause stress shielding effects particularly due to the difference in the stiffness of compact iron and bone. In order to avoid such effects, the manufacturing of porous iron structures has become a strategy to avoid problems related with the high stiffness of compact iron, and the low degradation rate. Iron porous structures with porosities in the interval 45.6–86.9% provide a compressive elastic modulus between 218–845 MPa [25]. These values are closer to the ones of the trabecular bone.

Bone is a porous tissue with dynamic rehabilitation in order to maintain its healthiness. The balance homeostasis of bone comes from the osteoclasts ability to reabsorb aged bone and the osteoblasts generation of new bone [6]. Bone is known for being the mechanical support of muscles and soft tissue, it is a load bearing hard tissue. When new stresses are applied to a bone it can causes bone homeostasis, remodelling the bone strength [6]. The insertion of metallic implants with a higher stiffness than the bone itself can cause stress shielding, bone reabsorption and the inhibition of bone formation. These consequences may lead to health complications, and possibly additional surgeries [6]. Topological design along with additive manufacturing produces porous structures that try to mirror the bone structure [15]. Among others, factors such as porosity, pore size and pore interconnectivity are key factors that will expressively influence the biological and mechanical properties of porous structures such as bone ingrowth and transportation of cells and nutrients [6]. Also the mechanical properties of the porous structures could be adjusted to mimic those of trabecular or cortical bone [19]. Although the focus of the present work is on iron porous structures, it is worth mentioning some features of compact iron, which may be used in load-bearing applications.

Heat treatments along with different manufacturing procedures may be used to change the mechanical properties of the iron structures [19,35]. Values of the mechanical properties of iron and iron-alloys are given in Table 2.

Biodegradable iron and iron-alloys tend to exhibit mechanical properties closer to others metals, such as 316L stainless steel [1,37]. When compared with other biodegradable metal such as Mg, Fe shows greater values of yield stress (YS) ultimate tensile stress (UTS) and fracture strain, ε_f_. The mechanical properties of biodegradable iron should be comparable with pure Ti, Ti6Al4V and 316L stainless steel, regarding to a load-bearing bone implant application [36]. Examples of the mechanical properties and degradation rate for iron and iron alloys that may be used in coronary stents are summarized in Table 3.

### 3.1. Influence of the Design

Topological design of porous metallic structures is used on medical structures to overcome problems from the stress shielding and inflammatory reactions caused by non-degradable implants.

Among the three-dimensional (3D) cellular materials, which includes foams, a novel type of 3D structure stemmed, designated as lattice structures. Lattice structures are repletion of unit cells with a well-defined geometry Lattice structured porous iron has been proposed with geometries as cubic, octahedral, pyramidal and diamond, also varying the porosity [21,25]. There is a wide variety of topologies being most of unit cells based on space-filling polyhedral [19].

Li et al. [29] presented a functionally graded porous iron study. Iron scaffolds were produced by selective laser melting (SLM), a powder bed-based technique from AM method. The porosity of the scaffolds evaluated by µCT were: S0.2 = 84.8 ± 0.1, Dense-in = 70.6 ± 0.4, Dense-out = 71.0 ± 0.2 and S0.4 = 58.4 ± 2.0. These abbreviations mean: “S0.2” is uniform strut thickness of 0.2 mm, “Dense in” is a strut thickness of 0.2 mm to 0.4 mm from the periphery to the center, “Dense out” is reversely thickness of 0.4 mm to 0.2 mm and “S0.4” is uniform strut thickness of 0.4 mm. Figure 3 illustrate the topological design of those scaffold studied [29]. Evaluating 4 different scaffolds designs, iron presented a smooth compress behavior, without fluctuations or sudden failures. Figure 4 shows the mechanical properties before and after degradation upon immersion in simulated body fluids [29]. Results showed that the mechanical properties are topologically dependent. AM has the possibility of fabricating topologically ordered porous metallic structures and these arrangements possess a fully interconnected porous structure, which mimic the bone’s mechanical properties.

Li et al. [41] studied cellular structures with a diamond unit cell with functional graded structures. Specimens also fabricated by selective laser melting (SLM) were tested throughout cytocompatibility and mechanical tests. These authors found that after 4 weeks of biodegradation in vitro, the structures were adequate for bone substitutes. The biodegradation mechanisms were found to be topology-dependent and different between the periphery and central parts of the structures [41].

Another work by Sharma and Pandey [42] studied the effect of pore morphology and porosity on the corrosion rate of biodegradable iron prostheses. The samples consisted of random porosity iron scaffolds and topologically ordered open cell iron scaffolds. For the topologically ordered scaffolds, three types of unit cells structures were used, cubic (C), truncated octahedron (TO) and pyramid (P), as illustrated in Figure 5. Table 4 presents the results of the electrochemical tests on the topologically ordered porous scaffold.

For the topologically ordered scaffolds, it seems the increased macro porosity allowed a free flow of SBF, reducing the corrosion rate. Highly interconnected porous structures could have a lower number of favourable sites for inducing pitting corrosion [42].

### 3.2. Porous Iron Alloys

One of the most used methods to increase the degradation of iron is alloying it with another element(s), such as, manganese, silicon, platinum, sulfur, carbon, phosphorus and palladium [31,38]. There are several works in the literature that study the effect of adding elements to iron [16,43,44,45].

For example, the addition of Mn to the iron resulted in porous Fe-30Mn alloy with mechanical properties close to the ones shown by bone [24]. In fact, the Young’s modulus was 32.47 ± 5.05 GPa, while the yield strength was 106.07 ± 8.13 MPa. A corrosion rate of 0.73 ± 0.22 mm/year was obtained, which is higher than the corrosion rate of pure iron [24]. In other work, the development of Fe-(30 wt%)Mn-(1–3 wt%)Ag alloys shows an increase in the shear stress and in the micro-hardness with the increase in the amount of silver, which was also accompanied by an increase in the relative density of the structures [10]. Silver particles were homogeneously distributed in the structure, giving rise to higher density of the alloys. Although the addition of 3 wt% Ag content allowed obtaining high density, strength and corrosion rate, the optimum cytotoxicity and the antibacterial activity was achieved by the alloy with 1 wt% Ag content [10].

In a different study, the incorporation of copper to Fe-35Mn alloys in the interval of 0 to 10% wt Cu was studied by Mandal et al. [11]. The addition of copper increased the degradation rate, as copper precipitated in the iron matrix, inducing local galvanic cells. The alloy also presented an excellent antimicrobial activity.

Considering the importance of having calcium and magnesium in the alloy composition to promote a fast bone regeneration, Hong et al. [32] were able to prepare Fe–Mn–Ca/Mg alloys, which exhibited an enhanced in vitro corrosion rate in comparison with pure iron.

The addition of second phase particles, for example of Mg_2_Si into the Fe matrix can also increase the degradation rate [9]. The mechanism and the degradation rate of the Fe/Mg_2_Si composites were found to depend on the distribution and size of the reinforcement particles [9].

The addition of Au and Ag to obtain Fe–Ag and Fe–Au composites was addressed by Huang et al. [12]. The increase in Ag and Au provokes an improvement in the mechanical strength being the compositions Fe–5 wt % Ag, Fe–2 wt % Au, and Fe–10 wt % Au, the ones that exhibit the best mechanical performance. The particles of silver and gold are second phases dispersed in the iron matrix, being effective in the increase of mechanical strength and forming a large number of galvanic corrosion sites, which accelerates the corrosion rate of iron matrix [12].

Wegener et al. [46] manufactured cellular implants by replications method with two distinct porosities using Fe0.6P alloy. The Young’s modulus was found to be a function of structural density, being close to the ones of trabecular bone (Table 5). Although the cellular structure permits to have a high internal surface, results reveal a slow corrosion rate [46]. Nevertheless, porous structures showed good biocompatibility.

Although several works can be found on iron alloying, there is not a clear chemical composition that enables to gather all the requisites for porous structures applied in the biomedical field.

## 4. Iron Alloys with Antibacterial Properties

To achieve an appropriate biodegradable iron-alloy implant for medical applications, it depends on the biodegradability of the elements. The elements alloyed with iron should enhance the degradation rate, have none or low toxicity, good cytocompatibility, the appropriate mechanical strength and elastic modulus [37,46].

Broadly, alloying elements such as Pd, Pt, W, C, S, Si and Ga increase the degradation rate of iron-based alloys [44]. Alloying elements also aim to produce antiferromagnetic Fe-alloys with compatibility to magnetic resonance imaging (MRI). The addition of gold, silver, tungsten, platinum and palladium have been found to improve the mechanical properties through the solid solution and second phase strengthening and accelerate the degradation rate of iron alloy [30,47,48,49]. Focusing in combing cytocompatibility, hemocompatibility, degradation rate and mechanical properties, the best element choice falls on Co, W, C or S [50]. Introducing Ca and Mg to FeMnSi alloys results in the formation of fine precipitates that induce an increase in the corrosion rate [45]. Another possibility to tune the mechanical and degradation properties is to use a composite material, achieved by the addition of Fe_2_O_3_, carbon nanotubes or calcium phosphates to a Fe matrix [49].

The implants in vivo often cause variation in the body fluid pH, from 7.4 to 4 (acidic) or to 9 (alkaline), causing electrochemical variations from the equilibrium state of the human body, and pathological infections may occur within [51]. Additionally, it is known that elements with large potential differences, such as Ag (+0.7966 V) and Au (+1.83 V) in comparison with Fe (−0.44 V) can enhance the corrosion rate of iron [12]. The Ag or Au ions act as a cathode element, intensifying the micro galvanic cells composed by Ag/Fe or Au/Fe corroding the anode (Fe) [12]. Besides, Ag is a biomaterial, with antibacterial and antiseptic properties which could be used to treat antibacterial and inflammatory bone diseases [52]. Therefore, combining Ag with Fe for medical applications could be crucial to significantly increase the corrosion of iron and reduce the diseases caused by bacterial infections.

Sotoudehbagha et al. [10] designed nano-structured Fe-Mn-Ag alloys with different percentages of Ag and studied the cytotoxicity, antibacterial activity, corrosion rate and mechanical properties. Considering the importance of evaluating the iron corrosion, electrochemical test was performed using a Hanks’ Balanced Salt Solution (HBSS) at 25 °C with potentiodynamic polarization. The corrosion potential (E_corr_/mV) and corrosion current density (*i_corr_*/µAcm^−2^) were determined by the Tafel curves and the corrosion rate was calculated using the following ASTM G59 equation:(1)$$CR=3.27\times 10^{3}\frac{i_{corr}EW}{\rho }$$

The corrosion rate (CR mm/year) is calculated based on the oxidation of Fe to Fe^2+^ equivalent weight, *EW* = 27.92 g/eq (gram equivalent). The corrosion behavior and mechanical properties are illustrated in Table 6. According to Sotoudehbagha et al. [10], the density of the alloys increases with the addition of Ag as well as the ultimate shear stress and microhardness. The authors also observed that a more homogeneous distribution of Ag with the increase of the percentage of Ag within the austenite matrix.

Furthermore, it was observed that Ag provide micro galvanic sites that increase the corrosion current density of the alloys. Although, the iron alloy with a higher percentage of Ag (Fe-30Mn-3Ag) have shown higher antibacterial rate against *S. aureus* and *E. coli* bacteria, lower cellular metabolic activity toward human umbilical vein endothelial cells (HUVEC) was observed when compared to Fe-30Mn-1Ag (Figure 6). For these reasons, the Fe-30Mn-1Ag was considered the biodegradable antibacterial alloy with the best properties [10].

Another study was conducted on Fe-Au and Fe-Ag alloys production by Huang et al. [12], to support its use in biomedical applications, specifically on stents applications. The alloys were produced by a powder metallurgical process and sintered using spark plasma. To mimic the body environment, and considering that iron degradation it is strongly affected by the presence of oxygen, the electrochemical measurements were performed with oxygen (2.8 to 3.2 mg^−1^) dissolved in Hank’s solution medium [12]. The presence of Ag and Au in the alloys revealed that the as-sintered iron-based materials presented much finer grains than that of as-cast pure iron. Additionally, it was identified by XRD that iron alloys were composed of two distinct phases (α-Fe and pure Ag phases) in Fe-Ag system, and α-Fe and Au phases in the Fe-Au. The same study revealed that occurs an improvement in the mechanical strength with addition of Ag and Au, especially for the Fe-5 wt% Ag which have exhibited the best mechanical properties (Table 7). The increase in the corrosion rate of the iron matrix observed in the alloys with Ag or Au was attributed to more uniform corrosion detected in the alloys when compared with pure iron (Table 7, Figure 7a and Figure 8).

Despite the improvement in the mechanical properties and degradation rate, it has demonstrated that no significant toxicity on the L-929 cells and human umbilical vein endothelial cells EA were observed. Considering the concerns related with the bio application, the hemolysis results of all the developed iron-based biomaterials were within the range of 5% (Figure 7b), which is the criteria range value for been considered a biomaterial with good hemocompatibility. Additionally, the amount of platelet adhered on the surface of as-sintered iron-based materials was lower than that of as-cast pure iron, and the morphology of platelets was kept smoothly spherical on the surface of all the developed iron materials.

The corrosion mechanism proposed by the authors was mainly electrochemical corrosion with galvanic cells formation, which accelerates the corrosion rate of pure iron, as illustrated in the schematic representation presented in Figure 8 [12]. Between the iron matrix and Au or Ag, the iron acts as an anode, the iron matrix was oxidized into iron ions. Dissolved oxygen consumed the electrons generated by the oxidation of the iron matrix [12]. Ag and Au’s solid phases alkalinized the surrounding solution, promoting the formation of iron hydroxide (Figure 8b). Due to the instability of ferrous hydroxide, it becomes ferric hydroxide with the dissolved oxygen from the solution [12]. With the penetration of the solution under the Au and Ag solid phases, uniform corrosion with microgalvanic corrosion couples occurs [12].

In the case of the FeMnSi–MgCa alloys, very large negative corrosion potentials (E_corr_/mV) were found, ranging between −727 and −667.9 mV, being this alloy easily corrodible [45].

In another study conducted by Mandal et al. [11] in which they have developed a novel Fe-Mn-Cu alloy by powder metallurgy, with copper (Cu). The selection of copper was considered in this study due to its antimicrobial properties and knowing that copper beyond the limit of solid solubility, it would precipitate in the iron matrix, enhancing the mechanical properties and increasing the degradation rate by local galvanic cells [11]. In this study, only the γ-austenite phase (FCC) was detected, even for the maximum percentage of copper used and no significant change in the iron grain size or precipitation were observed with the addition of Cu [11]. Table 8 presents the corrosion potential, corrosion current and corrosion rate obtained for the Fe-alloys immersed in with Hank’s solution at 37 °C [11].

As expected, the corrosion rate of iron alloys depends on the percentage of copper added. Until 3 wt% of copper in the iron matrix, the corrosion rate is reduced, because copper forms a solid solution with iron [11]. When the amount of copper exceeds the limit of solubility in iron, copper precipitates in the Fe matrix, this phenomena results in local micro-galvanic cells formation, which increases the alloy corrosion rate [11].

Antimicrobial tests were performed in the presence of *E. coli.* bacteria, to prove the bactericidal effect of copper present in the alloys. It is known that pathogenic microorganisms use iron as an essential cofactor in the metabolic pathway, and the addition of cooper to the iron matrix can reduce the bacteria adhesion [11]. Additionally, the cytocompatibility of Fe-Mn-Cu alloys was also evaluated in osteoblastic MG63 cells following ISO-10993:12 standard. The results presented no cytotoxic of the alloys studied and for cell viability essay the addition of copper in Fe-Mn alloys did not show a significant difference, as illustrated in Figure 9 [11].

## 5. Biodegradation Behavior and Biocompatibility of Iron

The degradation of metallic implants inside the body increases the ion content levels, causing cytotoxicity. To prevent cytotoxicity, the degradation rate needs to be controlled, and the absorption should occur at the same rate as the tissue is repaired [3,36], [53]. The main challenge for iron is raise degradation rate by accelerating the process. High energy grain boundaries in iron alloys and finer microstructures attend to enhance the corrosion rate [54,55]. Further methods have been used to control the degradation rate, one of them making porous scaffolds and other alloying iron with specific elements, as Pd, Mn, Ca, Ag, which promote iron corrosion and improve mechanical properties [54,56]. The main process for degradation of porous iron scaffold in SBF is diffusion process [54].

Grain refinement methods are currently for altering the way metals degrade. An advantage of these techniques is that the chemistry of the metal remains unchanged [55]. Fine grained metals have also been discovered to provoke a weaker inflammatory response in hosts, with an overall better interaction. Ralston and Birbilis have introduced a relationship between the materials average grain size and its corrosion rate, similar to Hall-Petch equation, given by [55]:Icorr = A + Bd^1/2^(2)
This equation represents the importance of the average grain size, d, to the corrosion current, I, since A is a constant value depending on the material purity and composition and B is also a constant depending on the media’ nature [55].

### 5.1. Corrosion Mechanism of Iron in Physiological Conditions

Body fluids are an aqueous aggressive environment that provokes electrochemical corrosion of metals. Controlling the corrosion rate helps controlling the cytotoxicity by metals, achieving a balance between the release rate of corrosion products and the ability of the body to absorb and excrete them [6]. The intimate link between degradation and mechanical integrity of the biodegradable iron implant is illustrated in Figure 10 [6,34]. Ideally, degradation begins at a very slow rate to maintain optimal mechanical integrity of the implant and increases at the same rate the body is healing itself. A period of 6–12 months is expected for the remodelling process to be completed [7]. It is important to emphasize that iron degradation should not be so fast that could cause an intolerable accumulation of degradation product around the implantation site. A total period of 12–24 months after implantation is considered reasonable for the stent to be totally degraded [7,34,53].

The corrosion rate is determined by kinetic factors and corrosion tendencies are determined by thermodynamic factors [57,58]. Metal corrosion in vivo is predominantly driven by chloride ions present in body fluids, namely the contact with blood and interstitial fluid. The chloride ion concentration in plasma is 113 mEq L^−1^ and in interstitial fluid is 117 mEq L^−1^, despite the low value is capable of corroding metallic implants [59]. In addition, chemicals such as amino acids and proteins found in body fluids tend to accelerate corrosion. The pH of the body fluid changes little acting as a buffer solution. Normal blood and interstitial fluids have a pH of around 7.35–7.45, although it can decrease near surface implantation areas and isoelectric points of biomolecules, such as proteins [59].

Recent research made by Sharma et al. [60] measured the pH increase in 28 days of additive manufactured porous iron in SBF solution. The results showed an increase in pH of 0.5 ± 0.05.

Li et al. [21] observed an increase of pH 7.4 to 7.8 after 28 days immersion in r-SBF solution for an iron scaffold with 80% porosity made by direct metal printing.

A general representation of metallic interfaces reacting with body fluid is present in Figure 11, where the metal reacts with the environment, release positive ions (M^n+^) to the environment, keeping electrons (e^−^) to the metal substrate. The contact of surface metal with body fluid results in oxidization of the metal to a more stable ion [61]. The reactions lead to formation of a protective metal oxide layer on the surface (yellow spots). The interactions with the body fluids may lead to deposition of calcium phosphate on the metal oxide layer, which permit that cells adhere on the surface to form tissues [61].

Metals with an electrode potential slightly higher than zero may, under certain environments inside the human body, be degraded [37]. Specific parameters, such as surface film condition and environmental aspects (e.g., pH and flow), influence the degree of corrosion kinetics and degradation process. These can be reflected by Pilling-Bedworth ratio and Pourbaix diagram [37].

The electrochemical corrosion of iron in physiological environment happens in an oxygen absorption mode, and can be expressed through the following reactions [56,62,63]:       Anodic reaction: Fe → Fe^2+^ + 2e^−^
(3)
            Cathodic reaction: O_2_ + 2H_2_O + 4e^−^ → 4OH^−^(4)
   Fe^2+^ +2OH^−^ → Fe(OH)_2_(5)
Fe^2+^ → Fe^3+^ + e^−^(6)
    Fe^3+^ + 3OH^−^ → Fe(OH)_3_(7)

Fe(OH)_3_ is hydrolysed in the presence of oxygen and chloride ions. Fe(OH)_2_ react with a part of FeO(OH), resulting in the formation of magnetite Fe_3_O_4_, a protective iron oxide layer, lowering the corrosion rate [56]:Fe(OH)_2_ + 2FeO(OH) → Fe_3_O_4_ + H_2_O (8)

Hank’s solution is composed by phosphates, sulphates, chlorides, and carbonates (see Table 9 and ref [51]). During anodic oxidation, Fe^2+^ ions may occur facilitating the formation of iron phosphate. The corrosion of pure iron in Hank’s solution increases the pH value, easing the precipitation and deposition of those phosphates. The proposed equilibrium equations are shown in Equations (9)–(16) [56]:    Fe(OH)_2_ + Cl^−^ → FeClOH + OH^−^
(9)
    FeClOH + H^+^ → Fe^2+^ + Cl^−^ + H_2_O (10)
     Fe^2+^ + O_2_ + 3OH^−^ → Fe(OH)_3_↓ + O^2^^−^(11)
     Fe(OH)_3_ + 2Cl^−^ → FeCl_2_OH + 2OH^−^(12)
     FeCl_2_OH + H^+^ → Fe^3+^ + 2Cl^−^ + H_2_O (13)
          6PO^3^^−^_4_ + 10Ca^2+^ + 2OH^−^ → Ca_10_(PO_4_)_6_(OH)_2_↓(14)
         3Fe^2+^ + 2PO^3^^−^_4_ + 8H_2_O → Fe_3_(PO_4_)_2_ 8H_2_O (15)
PO^3−^_4_ + Fe^3+^ → FePO_4_↓(16)

Generally, Pourbaix diagrams predict the stability and corrosion of metals in aqueous solution at 25 °C [37,51]. The diagrams also indicate regions of potential and pH in which the metal is protected from severe corrosion [37]. Moreover, diagrams provide the evidence of corrosion and the prediction of corrosion products. Figure 12 illustrates the calculated Pourbaix diagram for pure iron in physiological concentrations (37 °C). The concentration of HPO_4_^−2^ and HCO^−^_3_ (CO_2_(aq)) in the diagram are set to be identical to the concentrations in human blood plasma (0.001 mol L^−^^1^ for HPO_4_^−2^, and 0.027 mol L^−1^ for CO_2_(aq)) [37].

As foreseen on diagrams, iron in physiological conditions (T = 37 °C, pH = 7.4 and E = 0.78 V) will react and form solid Fe_2_O_3_. Furthermore, HPO_4_^−2^ will exist, but no carbonate ion species are expected to be present under the same conditions.

For comparison purposes Table 9 and Table 10 displayed the body fluids composition and the composition of various simulated body fluids, respectively.

As can be remarked the differences in ions concentrations are significant for different simulated body fluids and consequently influence the degradation behaviour of pure iron and the obtained corrosion products [2]. The interaction between pure iron and different kinds of ions and chemical species, as carbonates, chlorides is complex. The proposed equilibrium equations with HCO_3_^−^ are [51]:CO_2(g)_ ↔ CO_2(aq)_ + H_2_O → H_2_CO_3_ ↔ H^+^ + HCO_3_^−^(17)
6Fe + H_2_CO_3_^−^ + 12H_2_O ↔ Fe_6_(OH)_12_CO_3_ + 13H^+^ + 14e^−^(18)
6Fe^2+^ + HCO_3_^−^ + 12H_2_O ↔ Fe_6_(OH)_12_CO_3_ + 13H^+^ + 2e^−^(19)
6FeOH^+^ + HCO_3_^−^ + 6H_2_O ↔ Fe_6_(OH)_12_CO_3_ + 7H^+^ +2e^−^(20)
Fe_6_(OH)_12_CO_3_ ↔ 6α-FeOOH + CO_3_^2-^ + 6H^+^ + 4e^−^(21)
HCO_3_^−^ ↔ CO_3_^2−^ + H^+^(22)
Fe^2+^ + CO_3_^2−^ ↔ FeCO_3_
(23)
FeO + O_2_ → 2Fe_2_O_3_
(24)
FeO + H_2_O → Fe_3_O_4_ + H_2_(25)
7Fe(OH)_2_ + 2CO_3_^2−^ + H_2_O → 4Fe(OH)_2_∙2Fe(OH)_3_CO_3_ + H_2_ + 2OH(26)
4Fe(OH)_2_∙2Fe(OH)_3_CO_3_ + 4H^+^ → Fe_3_O_4_ + 3Fe^2+^ + CO_3_^2−^ + 8H_2_O(27)

The presence of chloride ions enhances the corrosion rate of iron and passivation is not accomplished. Chloride ions have a predominant presence to the corrosion process in comparison with bicarbonate and carbonate ions. Degradation products do not homogeneously cover surface, and chloride ions concentrates in preferential sites without the formation of a passive layer. The water hydrolyses the metal chloride, forming hydroxide and free acids, causing a local pH decrease. Corrosion pits growth wider and deeper following an autocatalytic reaction. Hydrogen carbonates and carbonates ions passivation effect is decreased by the chloride ions. Chloride ions inhibit the coalescence and crystallization of passive films, as well as the average thickness of oxide films. The final degradation product is predominantly γ-FeOOH. The proposed equilibrium equations with Cl^−^ can occur as [51,64]:Fe^2+^ + 2Cl^−^ ↔ FeCl_2_ + H_2_O ↔ Fe(OH)_2_ + HCl (28)
4Fe + Cl^−^ + 8H_2_O ↔ Fe_4_(OH)_8_Cl + 8H^+^ + 9e^−^(29)
4Fe2^+^ +Cl^−^ + 8H_2_O ↔ Fe_4_(OH)_8_Cl + 8H^+^ + e^−^(30)
4FeOH^+^ + Cl^−^ + 4H_2_O ↔ Fe_4_(OH)_8_Cl + 4H^+^ + e^−^(31)
Fe4(OH)_8_Cl ↔ 4γ-FeOOH + Cl^−^ + 4H^+^ + 3e^−^(32)
γ-FeOOH → Fe_3_O_4_(33)

Conversely, the interaction between phosphates species and iron species strongly depends on pH, concentration of dissolved oxygen and the concentration of phosphates and iron. The precipitation of fine iron phosphates is triggered with the FeOOH species surface, causing the adsorption of iron ions. Pitting is inhibited when the concentration of phosphate ions overlaps the chloride ions concentration. The proposed equilibrium equations with H2PO_4_^−^/HPO_4_^2−^ are [51,64]: 3Fe^2+^ + 2H_2_PO_4_^4−^ + 8H_2_O ↔ Fe_3_(PO_4_)_2_ + 2H^+^
(34)
Fe + 2H_2_PO_4_^−^ ↔ Fe(H_2_PO_4_)_2_ + 2e^−^(35)
3Fe(H_2_PO_4_)_2_ ↔ Fe_3_(PO_4_)_2_ + 4H_3_PO_4_(36)
3Fe + 2HPO_4_^2−^ ↔ Fe_3_(PO_4_)_2_ + 4H_3_PO_4_(37)
3Fe(H_2_PO_4_)_2_ ↔ Fe_3_(PO_4_)_2_ + 4H_3_PO_4_(38)
3Fe + 2HPO_4_^2−^ ↔ Fe_3_(PO_4_)_2_ + 2H^+^ + 6e^−^(39)
Fe3(PO_4_)_2_ + 6H_2_O ↔ 3γ-FeOOH + 2HPO_4_^2−^ + 7H^+^ + 3e^-^(40)
Fe_3_(PO_4_)_2_ + 4H_2_O ↔ Fe_3_O_4_ + 2HPO_4_^2−^ + 6H^+^ + 2e^−^(41)

### 5.2. In Vitro and In Vivo Biocompatibility

One of the first important studies about iron biocompatibility and degradation in vivo in coronary application was reported by Peuster et al. [65]. It was a one-year study in animal model, with pure iron and 316L stainless steel stents implanted in the aorta of pigs. Iron resulted to be a suitable metal for stent applications.

Zhang et al. [62] at a pioneer study on iron compatibility with blood and cell compares 99.9 wt% purity iron with magnesium-manganese-zinc alloy and 316L stainless steel in Hank’s solution. ISO 10993-4 standard was followed in this research and the haemolysis assay resulted in a low haemolysis ratio for iron. For the haemolysis assay, rabbit whole blood with 3.8 wt% sodium citrate was utilized. The prothrombin time assay resulted in excellent anticoagulant iron, and platelet adhesion tests in iron showed impressive anti-platelets adhesion. During cells toxicity, iron ions presented toxicity to the stem marrow cell of mouse bone. The standard ISO 10993-4 recommends the value of 5% in the haemolysis ratio to not cause haemolysis to blood system. Iron presented a ratio of 2.44%, accepted by the standard and proving iron has excellent anti-haemolysis property. Figure 13 illustrates the SEM images of platelet adhesion of iron, 316L stainless steel and Mg-alloy after 3 h immersed in rabbit blood plasma [62]. The number density of platelets on the surface for pure iron was 940 ± 164, for 316L stainless steel was 7211 ± 633 and for Mg-Mn-Zn alloy was 10270 ± 918. This analysis shows the iron anti-platelet adhesion property. The electrochemical behavior of pure iron was studied by open circuit potential-time tests, to measure the biocorrosion properties. The parameters obtained were E_corr_ = −0.510 V, I_corr_ = 1.68 × 10^−5^ A, E_b_ (break potential) = −0.40 V. The surface crystalline structure evaluated by XDR present phosphates as the main corrosion products, Mg_3_(PO_4_)_2_, Ca_3_(PO_4_)_2_ and Fe_3_(PO_4_)_2_·8H_2_O. Also, XPS spectra showed more Fe^2+^ compared to Fe^3+^. The increase of corrosion rate is time-dependent until a certain accumulation of products on the surface of iron. However no passivation stage was found at the Hank’s solution and no noble breakdown, possibly because the surface had not attained a protective film [62].

Iron ions may produce reactive oxygen species in cells. Highly reactive oxygen species can react with the most molecules found in cells, making them toxic. Furthermore, free iron can react with unsaturated fatty acids, resulting in the formation of lipid hydroperoxides and subsequently alkoxyl and peroxyl radicals. These products are capable of causing cell death and impair cellular integrity. Despite these oxygen specimens are damaging, they are normally generated in reactions and the body has defensive strategies against it. However, iron level needs to be limited in cells. The iron concentration should be less than 0.075 mg/mL [62].

Zhu et al. [66] assessed the biocompatibility of pure iron and cytotoxicity on endothelial cells was performed in SBF solution for one month at 37 °C. The incubation time was almost 700 h, the degradation rate was at the highest 40 µg/(cm^2^ h) and the mean rate was 20.4 µg/(cm^2^ h). The corrosion was predominately uniform corrosion. Endothelial cells from human umbilical vein were cultured with 10% fetal bovine serum, penicillin and streptomycin. The cells were incubated with iron solution for three days, with concentrations varying from 0 until 2000 µg/mL. The accessed cell proliferation show non-toxicity until 50 µg/mL iron concentration [66].

According to the literature [67], the grain size and texture considerably affect interactions between cells and osteoblast functions and roughness also potentially influences cell growth. The attachment, orientation, migration and metabolism of the human cells are determined by the properties of the metallic implant of austenitic stainless steel. The roughness of the grains is an important factor with regard to osteoblast adhesion and protein adsorption. Proteins and focal adhesion points of cells also interact at a scale that enables them to activate signalling pathways within the cell. These pathways, in turn, have an impact on the lifespan of the cell. Increased cellular activity is implied by an increase in cell attachment and pre-osteoblast proliferation, as well as a stronger presence of fibronectin. This is turn is linked to the physico-chemical properties of the surface of the metallic implant. The attachment of cells to the implant and their growth on its surface—and thus the compatibility between them—is influenced by the chemical and morphological properties of the surface. Hydrophilicity, ionic bonding, electrostatic and van der Waals interactions are the most significant factors that drive adsorption of macromolecules and proliferation of cells on the implant surfaces. Protein adsorption, cell spreading, and cell proliferation may be assisted with high surface energy and high surface hydrophilicity and wettability. Those characteristics are controlled by the grain size of the metallic implant [67].

Another study [68] compared pure iron and cobalt chromium coronary stents in vivo in domestic pigs for 28 days. The morphometric comparison between these two types of coronaries resulted in less inflammation of iron compared to cobalt chromium, adding the vantage of iron being radio-opaque, while cobalt chromium is radiolucent. The stents did not cause harm to the vessel, no peripheral embolization or thrombosis were found with the angiography. Although the degradation of the iron coronary stents has not been evaluated, it is noticed the brown coloration of the tissue around the stent. A possible reason is the assimilation of iron salts by the tissue. The inflammation caused by iron stents and its degradation products was not worse than the inflammation caused by the cobalt chromium stent [68].

A 36 month study was presented about the degradation, absorption and biocompatibility of a nitrided iron (Fe alloyed with 0.074 wt% N) coronary stent with 70 µm height [69]. The device was compared with other stents made by PLLA-based, magnesium based, Co-Cr, pure iron scaffold and stainless steel. For in vitro corrosion tests, phosphate buffered saline (PBS) with pH at 7.4, flux speed at 25 ± 5 cm/s, oxygen at 4 ± 0.5 mg/L and temperature of 37 °C was used to simulate the inner environment of a coronary artery. For in vivo experiments, stents made by nitrited iron, pure iron and 316L stainless steel were implanted at the abdominal aortas of rabbits. Also, nitrited iron stents were allocated at the coronary artery of minipigs (porcines). Pure iron has lower mechanical strength in comparison with 316L stainless steel and Co-Cr alloy, alloys traditionally used for manufacturing permanent stents. Iron can stay implanted for 18 months, therefore the balance between strength, ductility and biodegradation is the key to accomplish a suitable iron stent. Some alloys can increase the corrosion rate and mechanical performance, but they compromise the cytocompatibility, so the element needs to be in lower concentrations. As Fe-Mn alloy the in vivo corrosion of nitrided iron stent, the results presented a higher corrosion rate for nitrited iron in comparison with pure iron. After 12 months the mass loss of nitrited iron was 44.5 ± 6.4 wt% while the mass loss for pure iron stent was 24.0 ± 5.6 wt%. The initial radial strength of nitrited iron stent was 171 ± 5 kPa, after six months of in vivo performance was proximately 150 kPa and after nine months it was above 120 kPa [69]. The device performance of a coronary stent in terms of foreshortening, recoiling and crossing profile should be minimum. Side-branch accessibility and expansion diameter should be maximum as possible. Mechanical properties as radial strength should be between 110 and 170 kPa, high enough to support vessels lesion but still flexible, as clinically tested for stents Φ 3.0 × 18 mm. The studied nitrite iron stents presented better performance in comparison with the stents already in the market, such as, the Co-Cr alloy stent, the Mg-based stent and polymer-based stents [69]. Corrosion products were identified by XPS and the results showed strong signals of C, O, Ca, P and Fe presented a binding energy of 709.8 eV and Fe^3+^ binding energy was 712.31 eV. The degradation products identified by Raman analysis were Fe3O4, α-Fe2O3 and γ-FeOOH. The endothelialisation assess of nitrited iron stent was compared to a peer of 316L stainless steel after seven days inside a rabbit abdominal aorta, analysing the neointima coverage extend. Coronary stents are prejudicial to the endothelium, which is formed by a single layer at the vascular wall of endothelial cells [69]. The damaged at the endothelium lead to neointimal hyperplasia and may lead to stent thrombosis [70]. The analysis of endothelialisation was preformed to determine the neointimal hyperplasia 7 days after implantation, as illustrated in Figure 14.

A homogeneous endothelium layer was formed with the nitride iron coronary, lowering the risk of stent thrombosis. The 316L stainless steel coronary seemed to interrupt the endothelium natural recover [69]. The follow-up local tissue response for the minipigs and the rabbits, 53 months and 36 months, respectively, showed no pathologic changes or abnormalities of the organs. After 53 months of nitrited iron stent implementation, the images by Micro-CT 2D presented a non-uniform degradation and absorption inside the porcine coronary. The corrosion products presented a moving tendency from *in situ* and peri-stent areas to tunica externa, also known as tunica adventitia, the outermost layer of the blood vessel. To essay the biosorption of the corrosion products, the in vivo analysis of the stents in minipigs was chosen because porcine coronary artery is closer to human coronary artery and the life time of a minipig is higher than a rabbit. At body fluid environment with pH 7.4, it is difficult to dissolve Fe_3_O_4_, Fe_3_(PO_4_)_2_, Fe_2_O_3_, Fe(OH)_3_ and FeOOH. Following the Pourbaix diagram of iron corrosion at a pH of 7.4 and phosphate physiological environment, Fe(OH)_3_, FeOOH, Fe_2_O_3_ (non-magnetic) present a steady state. Other types of corrosion products with lower stability is Fe_3_(PO_4_)_2_ and Fe_3_O_4_, because of their slow reaction kinetics. The natural organism low concentration of iron ions could make these corrosion products easily absorbed, since the solubility equilibrium convey towards the concentration of iron ions. The bioresorption of hydroxides and ferric oxides in body solubility, is slow and long-term. The insoluble products, resulted from iron corrosion, could take five to six years to complete bioresorption [69].

Li et al. [21] were the first authors to report a study on the topological ordered porous iron made by direct metal printing, using a diamond unit cell. Evaluated the in vitro corrosion of iron scaffolds in r-SBF at 37 °C for 28 days. Figure 15 represents the iron corrosion and weight loss after 28 days. The iron scaffold had a weight reduction of 3.1% after sample cleaning. 

The corrosion products’ morphology and composition showed iron carbonate (FeCO_3_) and iron protoxide (FeO) and FTIR also presented hydroxides and phosphates. Scanning electron microscope (SEM) analysis of the external structure, Figure 16, reveal a white layer at the surface only after 1 day of immersion. From day 7, the structure surface presented shiny white loose degradation products, and after 28 days, these degradation products covered surface structure almost completely. The scaffold geometry interfered with the degradation at the center and at the periphery region. The degradation products at day 7 were thinner and more condensed at the centre. At the periphery, the degradations products were loose and thicker. The periphery region presented more phosphorus and calcium [21].

Spot 1, spherically shaped and containing C, O, Ca and FeSpot 2, feather shaped and containing C, O and Fe.

The mechanical properties of AM iron remain similar to the properties of trabecular bone even after 28 days of biodegradation, which is an advantage compared to other metals. The degradation rate of the topological scaffolds was found to be 12 times larger than the one of compact iron. A suitable cytocompatibility was also observed. Another mechanical requirement for an orthopaedic material that lasts from weeks to one year is to have a strain in the interval 1.1 to 2.1, [1], which was adequate [21].

Moreover, in vitro cytocompatibility and degradation of iron porous scaffolds obtained using Fe-30Mn powder and binder jet printing point out a significantly faster degradation rate of the Fe-30Mn alloy compared to pure iron [24].

Li et al. [41] studied the corrosion caused by fatigue of iron ordered scaffolds, produced by selective laser melting (SLM), finding a revised simulated body fluid with cyclic load and degradation. One of the obligations of bone substituting biomaterials is the mechanical loading resistance, therefore with a good resistance to fatigue fracture. 70% of yield stress in air and 65% of yield stress in r-SBF, mainly due to iron’s ductility and slow degradation. Cyclic loading increased the degradation rate of iron, but a high fatigue strength remains, among others mechanical and chemical properties. This study present iron as a suitable bioactive bone implant [41].

### 5.3. Influence of Surface Treatments on Biodegradation Behaviour

Surface treatments are used primarily to enhance the biocompatibility of iron, increase the corrosion rate, and also aimed for an uniform corrosion of iron. The surface functionalization of iron to improve its biocompatibility is reported in studies including ion implantation with tantalum, with lanthanum, formation of Fe-O film, plasma nitriding, coatings with calcium phosphates and polymeric coatings [1,48,71]. Cytocompatibility and osseointegration are bioactivities improved by coating the metallic surface [56].

Huang and Zheng [48] measured pure iron degradation by coating with platinum (Pt) discs arrayed in pattern by photolithography and evaporation by electron beam. The patterned was adopted to control the degradation rate and regulate cells proliferation and adhesion. Platinum was chosen for its hemocompatibility and high corrosion potential, forming galvanic cells with pure iron. Platinum presents cytotoxicity but the chemical stability prevents for ions released into the body. The platinum discs had two designs, one with 20 µm diameter, the nearest space between discs was 5 µm and the thickness was approximately 285 nm (Φ20 µm × S5 µm). The second design was Φ4 µm × S4 µm with thickness around 80 nm [48]. Electrochemical corrosion tests in Hank’s solution demonstrated significantly increase the corrosion current density (I_corr_) and decrease of corrosion potential (E_corr_) as listed on Table 11, that also present the corrosion rate in static immersion for 42 days. The higher degradation of coated iron in comparison with pure iron is a consequence of galvanic cells formed between platinum discs and iron matrix [48].

Platinum discs improved degradation rate and biocompatibility of pure iron, as the results of cytotoxicity tests with cell viabilities, human umbilical vein endothelial cells and human vascular smooth muscle cells. Haemolysis and platelet adhesion, followed by ASTM F756-08, was analyzed and the risk of thrombosis caused by pure iron can be decreased by the platelet adhesion on Pt coated Figure 17 illustrates the hemolysis percentage and number of adhered platelets found in this study [48].

Aiming the improvement of iron biocompatibility, surface modification with hydroxyapatite as a coating presents great outcomes. Hydroxyapatite is widely used as coating for metallic protheses, since hydroxyapatite is the main mineral constituent of bone and presents outstanding bone integration [23]. The in vivo osteointegration and the cytocompatibility of the protheses are affected by the bonding between coating and substrate and the morphology of the coating. The morphology is expected to be homogenous and uniform, also with high strength bonding and most of all the coating must improve biocompatibility. Nano-Plotter 3D printing iron scaffolds with tailored mechanical behaviour and were coated with nanostructured hydroxyapatite by hydrothermal method, as illustrated in Figure 18 [23].

The coating successfully reduced the release of Fe ions, to below 2 mg/L for 120 µm hydroxyapatite thickness, increasing the cytocompatibility of rabbit bone marrow mesenchymal stem cells (rBMSCs). The study compared the osteogenic differentiation with the analysis of alkaline phosphatase in scaffolds with and without hydroxyapatite coating. The results showed an increased activity of alkaline phosphatase (ALP) activity of rBMSCs with the hydroxyapatite coating, indicating a osteogenic bioactivity of AM iron scaffolds [23].

Another perspective about the coated porous structures is the improvement of antibacterial coating response compared to solid implants, partly because of a higher specific surface area [72]. Iron foams were coated with polyethyleneimine (PEI), an organic polymer with biological applications which was also used to enhance the cytocompatibility of iron and its degradability [13].

Hong et al. fabricated by binder-jet 3D printing alloys with Fe-35 wt% Mn and Fe-34 wt% Mn-1 wt% Ca, evaluating the cytocompatibility and degradation behaviour [32]. The mechanical properties tests were conducted following ASTM E8-04 and the electrochemical corrosion tests were performed with Hank’s solution HBSS H1387, at 37.4 °C. The corrosion current density I_corr_ and the corrosion potential E_corr_ were determined by Tafel analysis from potentiodynamic polarization curves, as illustrated in Figure 19 [32].

The calculate corrosion current density increase as the alloy elements Ca and Mg concentrations rises in the alloy and corrosion potential follows a more negative value trend as Ca and Mg concentrations rises (Table 12).

The cell viability of the 3D Fe-alloys, as presented in Figure 20, were evaluated by murine osteoblast-like MC3T3-E1 cell line, at 37.4 °C for 72 h, with live cells as green light. The analysis suggested better cytocompatibility for Fe-Mn-1Ca and Fe-Mn-2Ca in comparison with the alloys incorporating magnesium [32].

## 6. Conclusions and Future Trends

Biodegradable iron is a topic that has been under discussion for some time, combined, for example, with the possibility of creating CT-images of metallic absorbable implants. Biodegradable iron has gained new importance due to new production techniques, which allow the development of porous structures with controlled porosity. Custom-made porous structures of iron obtained with new manufacturing processes, such as additive manufacturing, will permit the tailoring of mechanical and corrosion properties, which is a substantial advance in tissue engineering for biomedical applications and broadly in the field of materials engineering. Successful applications of biodegradable iron are expected in the tissue engineering field, with a particular emphasis on temporary implants in non-load bearing orthopaedic applications and cardiovascular devices.

This review points out several strategies to overcome the drawbacks of pure iron, such as the use of porosity and the alloying of selected elements to simultaneously increase the iron corrosion rate and lower the stiffness. Still, several challenges will have to be overcome. The in vivo studies, which are rare, will make a step forward towards the clinical usage of this type of materials. Research gaps have been identified, and future directions have been pointed out in addressing the challenges for the development of iron-based alloys and composites to serve as biodegradable medical devices. In fact, future research and the development of biodegradable iron tends to move towards ‘‘multifunctional capabilities”. This means that there will be a tendency towards combining several aspects such as porosity, alloying of selected elements, composites and coatings to promote structures with a direct and active interaction with the host.

## Figures and Tables

**Figure 1 jfb-13-00072-f001:**
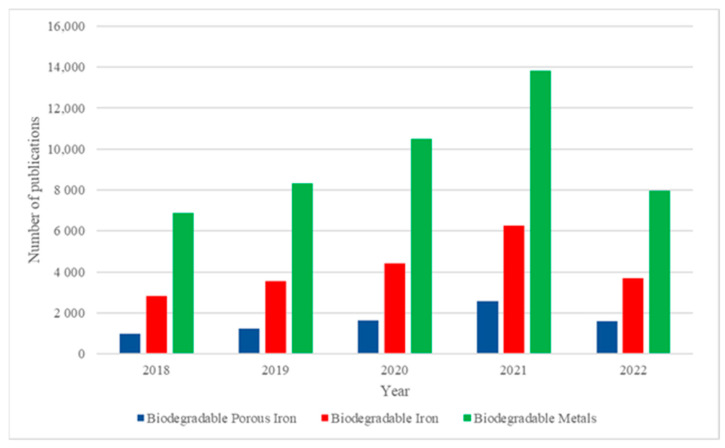
Number of publications related to biodegradable metals, iron and porous iron from 2018 until 2022 according to science direct database (data obtained in 17th of April of 2022). A steady increase in the number of publications indicates growing interest in the field of biodegradable iron biomaterials.

**Figure 2 jfb-13-00072-f002:**
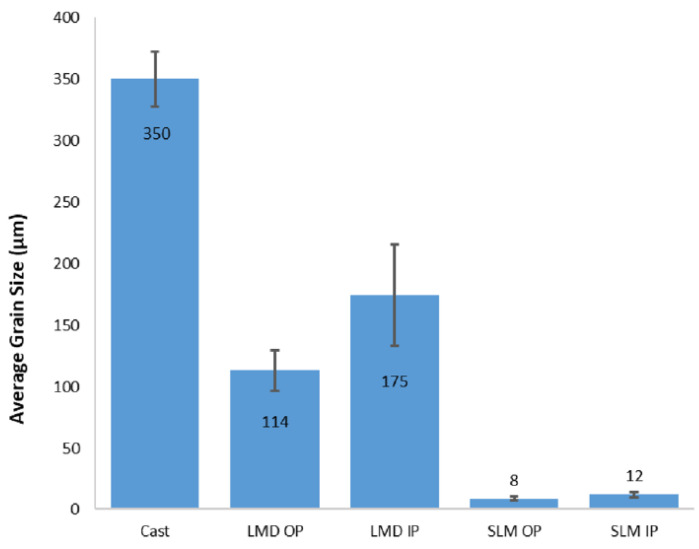
Average grain size in µm of LDM, SLM and casting pure iron (Reprinted with permission from Ref. [26]. Copyright @ 2019, Wiley Online Library).

**Figure 3 jfb-13-00072-f003:**
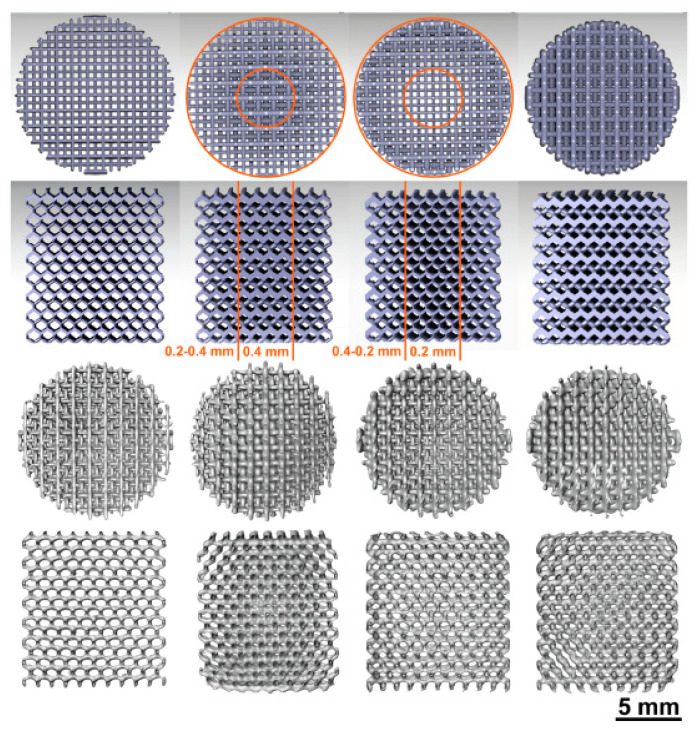
Illustrations of topological designs. Left to right: S0.2, Dense-in, Dense-out, S0.4. Top to bottom: top view and longitudinal cross-section obtained from CAD models, top view and longitudinal cross-section of the micro-CT reconstructions of the AM porous individuals (Reprinted with permission from Ref. [29]. Copyright @ 2019, Elsevier).

**Figure 4 jfb-13-00072-f004:**
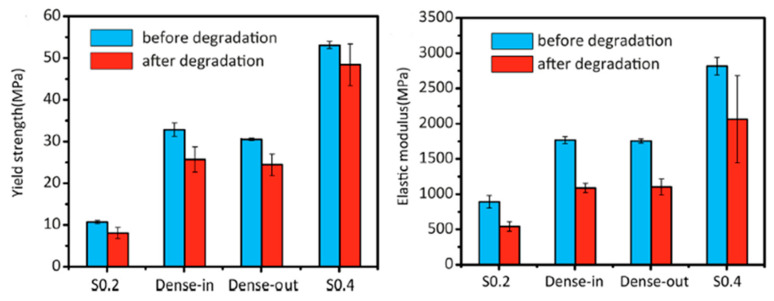
Mechanical properties of iron scaffolds comparison of before and after 28 days of degradation. (Reprinted with permission from Ref. [29]. Copyright @ 2019, Elsevier).

**Figure 5 jfb-13-00072-f005:**
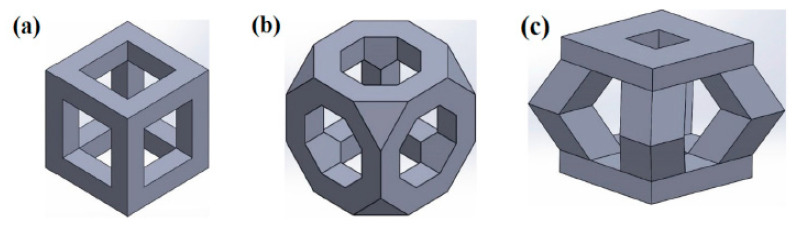
3D printing CAD models for topologically ordered porous iron structures, (**a**) cubic, (**b**) truncated octahedron, (**c**) pyramid individuals (Reprinted with permission from Ref. [42]. Copyright @ 2019, Elsevier).

**Figure 6 jfb-13-00072-f006:**
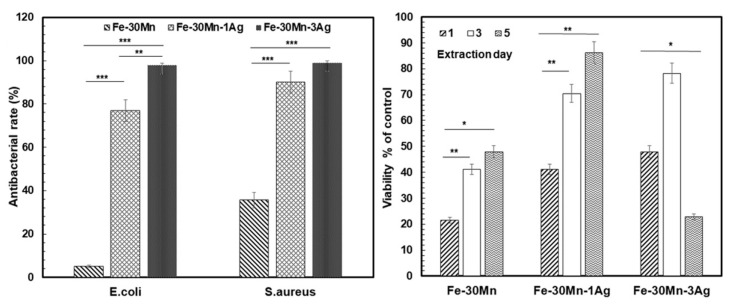
Antibacterial rate and cell viability of Fe-30Mn, Fe-30Mn-1Ag and Fe-30Mn-3Ag (*** is *p* < 0.001, ** represents *p* < 0.01 and * is *p* < 0.05) (Reprinted with permission from Ref. [10]. Copyright @ 2018, Elsevier).

**Figure 7 jfb-13-00072-f007:**
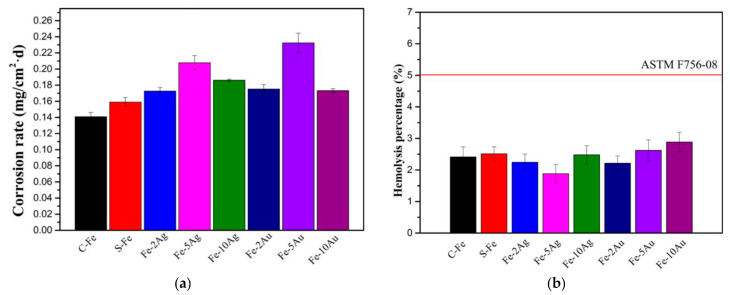
(**a**) Corrosion rate after 30 days in dynamic immersion in Hank’s solution and (**b**) hemolysis percentage of the iron-based materials (Reprinted with permission from Ref. [12]. Copyright @ 2015, Wiley Online Library).

**Figure 8 jfb-13-00072-f008:**
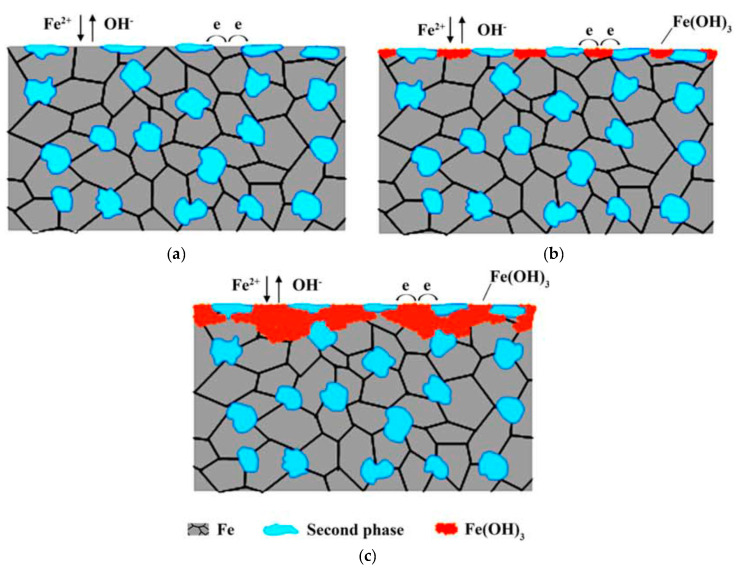
Schematic illustration of corrosion mechanism for Fe-Au and Fe-Ag: (**a**) corrosion reaction initially started, (**b**) hydroxide layer firstly appearance, (**c**) formation of hydroxide layer (Reprinted with permission from Ref. [12]. Copyright @ 2015, Wiley Online Library).

**Figure 9 jfb-13-00072-f009:**
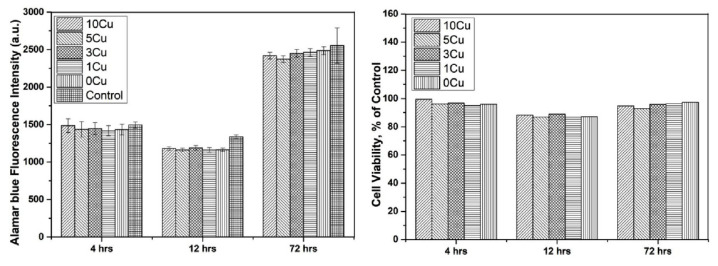
Alamar blue and cell viability of MG63 for Fe-Mn-Cu alloys (Reprinted with permission from Ref. [11]. Copyright @ 2019, Elsevier).

**Figure 10 jfb-13-00072-f010:**
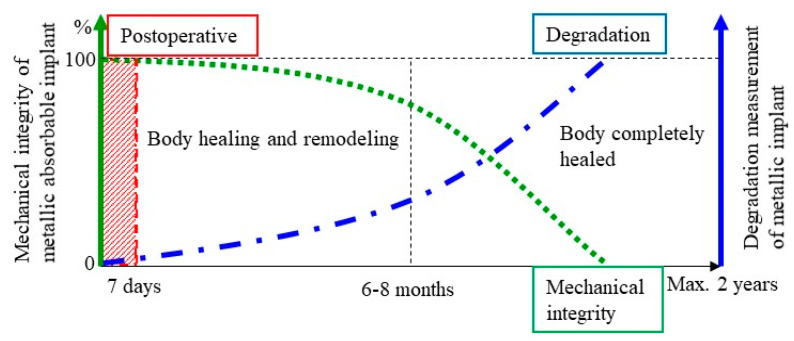
Illustration of the ideal compromise for absorbable metals in coronary stent application. Degradation rate stays low until 6–8 months, and mechanical integrity stays high. For absorbable bone implant, a similar illustration is valid, with a lower mechanical integrity during 3–6 months. (Adapted/Reprinted with permission from Refs. [3,36]. Copyright @ 2014, Elsevier).

**Figure 11 jfb-13-00072-f011:**
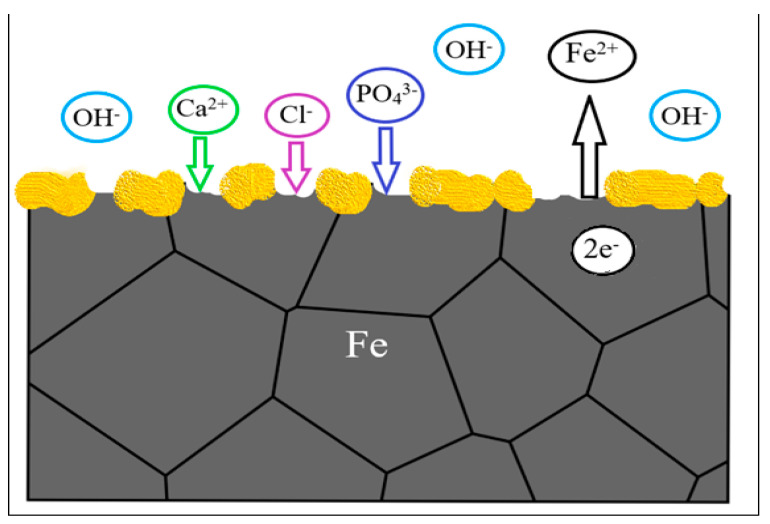
Diagram of metal interface degradation in physiological medium.

**Figure 12 jfb-13-00072-f012:**
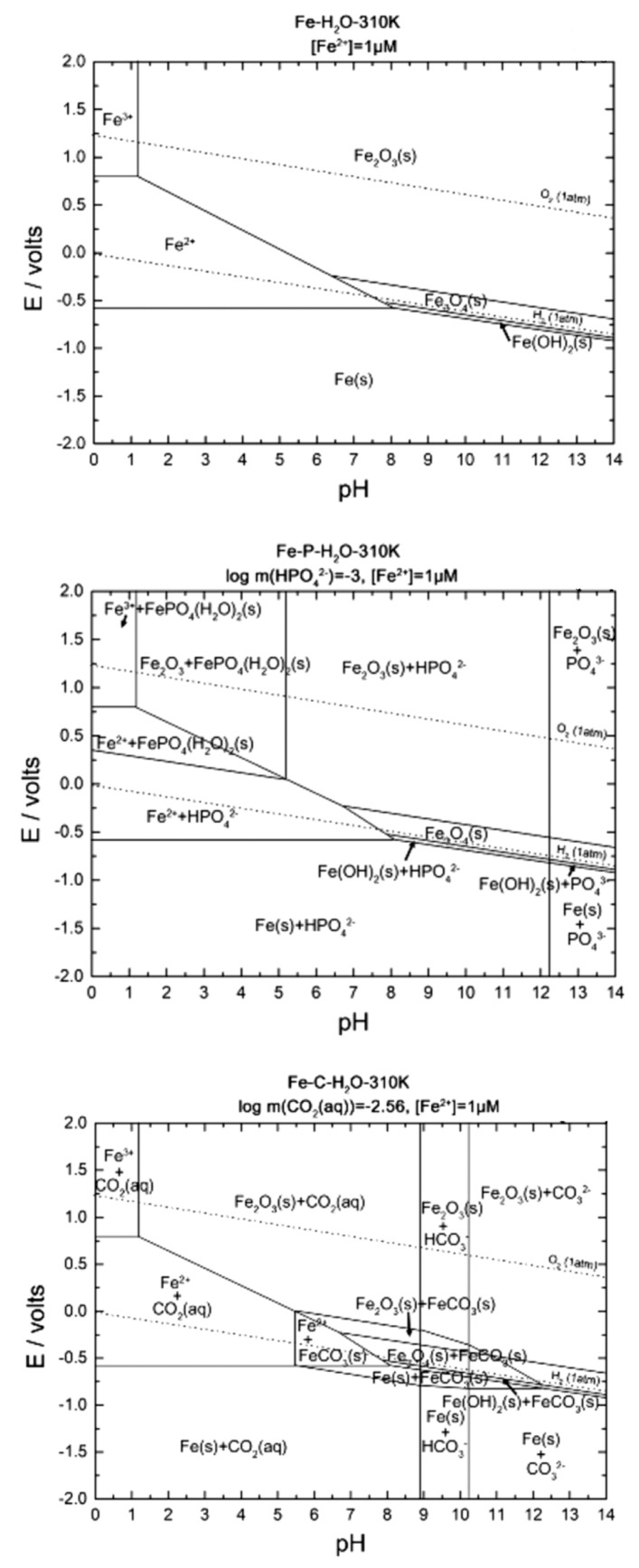
Pourbaix diagrams for Fe in physiological concentration and body temperature: Fe-H2O diagram, Fe-P-H2O diagram and Fe-C-H2O diagram, respectively (Adapted/Reprinted with permission from Ref. [37]. Copyright @ 2019, Wiley Online Library).

**Figure 13 jfb-13-00072-f013:**
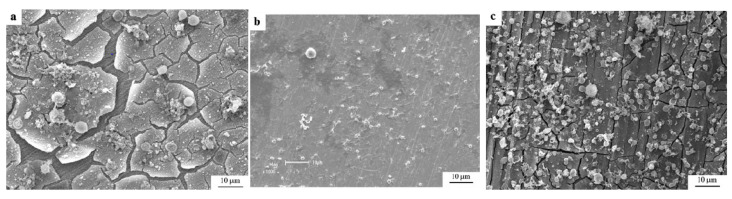
SEM images of platelets adhesion in blood plasma during 3 h: (**a**) pure iron, (**b**) 316L stainless steel, (**c**) Mg-Mn-Zn alloy (Reprinted with permission from Ref. [62]. Copyright @ 2010, Springer).

**Figure 14 jfb-13-00072-f014:**
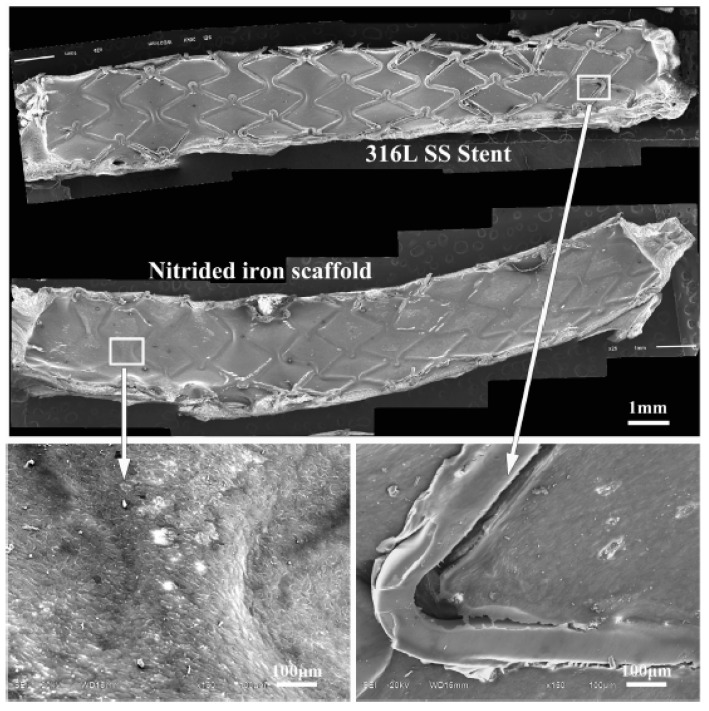
Endothelialisation of rabbit abdominal aorta with nitrided iron coronary stent and 316L stainless steel stent (Reprinted with permission from Ref. [69]. Copyright @ 2017, Elsevier).

**Figure 15 jfb-13-00072-f015:**
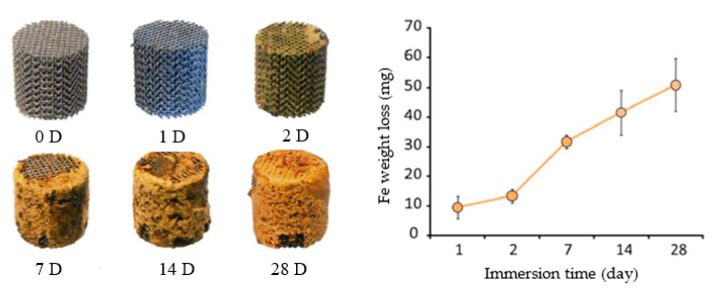
In vitro corrosion products and weight loss of AM porous Fe samples (Reprinted with permission from Ref. [21]. Copyright @ 2018, Elsevier).

**Figure 16 jfb-13-00072-f016:**
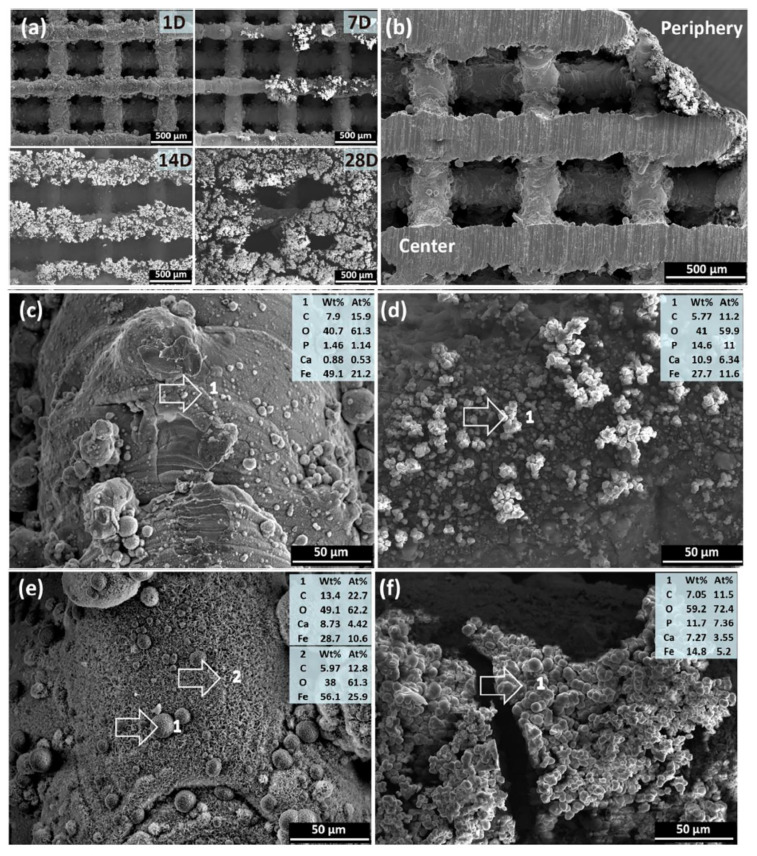
SEM and EDS analysis of degradation products from the scaffold periphery to the centre; (**a**) degradation with 1 day, 7 days, 14 days, and 28 days. (**b**) scaffold cross-section after 7 days immersed. (**c**,**d**) 7 days degradation on the center and on the periphery, respectively. (**e**,**f**) 28 days degradation on the center and on the periphery, respectively. EDS analysis was performed on: Spot 1, spherically shaped and containing C, O, Ca and Fe; and: Spot 2, feather shaped and containing C, O and Fe (Reprinted with permission from Ref. [21]. Copyright @ 2018, Elsevier).

**Figure 17 jfb-13-00072-f017:**
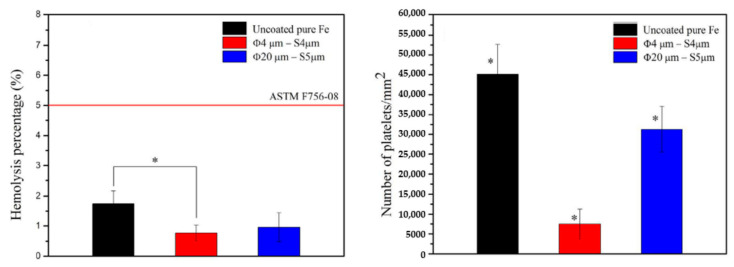
Haemolysis percentage and number of adhered platelets on pure iron and on iron coated with platinum discs. * represents *p* < 0.05 (Reprinted with permission from Ref. [48]. Copyright @ 2016, Elsevier).

**Figure 18 jfb-13-00072-f018:**
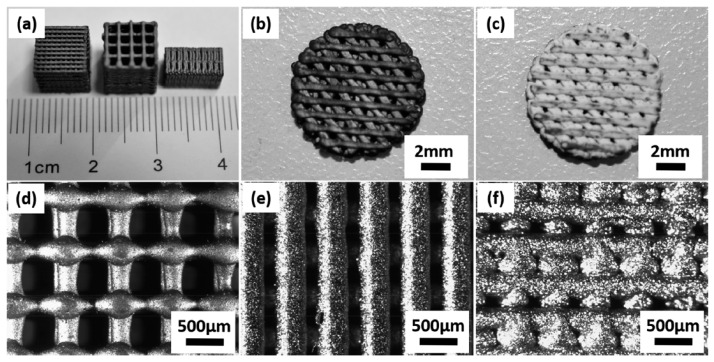
Iron scaffolds produced by 3D printing: (**a**) variety of designs applied, (**b**) in vitro study without hydroxyapatite coating, (**c**) in vitro study with 4 layers of hydroxyapatite coating, (**d**,**e**) top view of scaffolds, (**f**) side view of iron scaffolds (Reprinted with permission from Ref. [23]. Copyright @ 2018, ACS).

**Figure 19 jfb-13-00072-f019:**
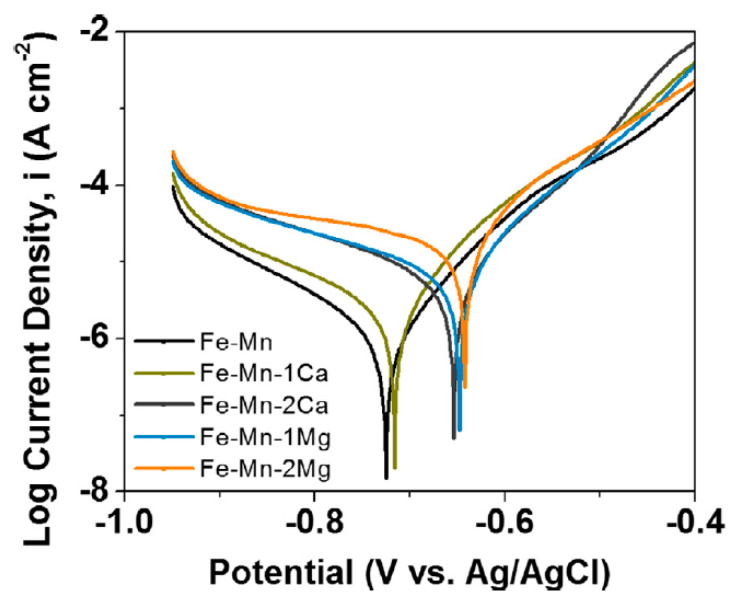
Potentiodynamic polarization of sintered Fe-Mn, Fe-Mn-Ca and Fe-Mn-Mg (Reprinted with permission from Ref. [32]. Copyright @ 2016, Elsevier).

**Figure 20 jfb-13-00072-f020:**
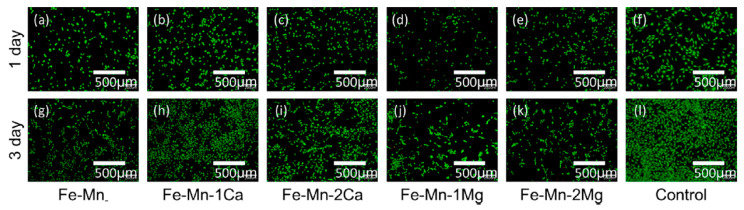
Cell viability assay with MC3T3-E1 cells for different binder-jet 3D Fe-alloying during three days at 37.4 °C, with fluorescent images of live (green) cells (Reprinted with permission from Ref. [32]. Copyright @ 2016, Elsevier).

**Table 1 jfb-13-00072-t001:** Composition of pure iron used for manufacturing SLM, LMD and casting in wt.% (Reprinted with permission from Ref. [26]. Copyright @ 2019, Wiley Online Library).

Sample	Density ρ (g/cm³)	Young’s Modulus (GPa)	Compressive Strength at 20% Strain (MPa)
SLM	7.87	199.7 ± 6.7	760.2 ± 6.5
LMD	7.84	202.5 ± 5.3	580.6 ± 4.2
Cast	7.81	202.5 ± 6.7	497.8 ± 7.5

**Table 2 jfb-13-00072-t002:** Mechanical properties and corrosion rate of iron and iron alloys (Adapted with permission from Ref. [36]. Copyright @ 2014, Elsevier).

Fe and Alloys	Yield Strength (MPa)	Tensile Strength (MPa)	Maximum Elongation (%)	Corrosion Rate * (mm/year)
Young’s modulus—200 GPa, density—7.8 g/cm^3^				
Pure iron as annealed	150	200	40	0.1
Fe-21Mn-0.7C as recrystallized	345	980	62	0.13
Fe-21Mn-0.7C-1Pd as recrystallized	360	970	64	0.21
Fe-10Mn forged	650	1300	14	7.17
Fe-10Mn-1Pd forged	850	1450	11	25.10
316L SS	190	490	40	-

* Corrosion rate data were collected from those having the most similar experiments, i.e., in simulated body fluid at 37 °C using polarization test, but they may not be directly comparable due to possible variation in specific testing condition and parameters.

**Table 3 jfb-13-00072-t003:** Biodegradable iron coronary stents mechanical and corrosion properties.

Iron and Iron Alloys	Average Grain Size (µm)	Yield Strength (MPa)	Tensile Strength (MPa)	Elongation (%)	In Vitro Corr. Rate (mm/year)
Armco® Fe annealed [38]	12–30	150	200	40	0.19
Fe-35Mn alloy annealed [31]	<100	230	1450–1550	30	0.44
Fe-30Mn-6Si solution treated [39]	<100	180	450	16	0.30
Fe electroformed and annealed at 550 °C [40]	2–8	270	290	18	0.46–1.22

**Table 4 jfb-13-00072-t004:** Porosity and corrosion parameters of iron topologically ordered scaffolds evaluated in SBF (Reprinted with permission from Ref. [42]. Copyright @ 2019, Elsevier).

Topologically Ordered Porous	Porosity %	Icorr (µA/cm²)	Ecorr (mV)	Experimental CR (mmpy)
Cubic 1 mm strut	86.90	62.95 ± 1.52	−684.7 ± 15.4	0.625 ± 0.02
Cubic 1.25 mm strut	71.80	87.75 ± 1.40	−691.2 ± 19.1	0.795 ± 0.02
Cubic 1.5 mm strut	50.70	170.00 ± 4.13	−706.5 ± 28.4	1.474 ± 0.10
Trunc. Oct. 1 mm strut	80.97	73.34 ± 2.01	−668.6 ± 18.0	0.710 ± 0.04
Trunc. Oct. 1.25 mm strut	59.99	115.87 ± 2.51	−690.7 ± 24.3	1.031 ± 0.02
Trunc. Oct. 1.5 mm strut	45.63	193.34 ± 3.22	−759.6 ± 31.1	1.640 ± 0.04
Pyramid 1.5 mm strut	54.82	135.10 ± 2.20	−747.1 ± 21.2	1.191 ± 0.03

**Table 5 jfb-13-00072-t005:** Mechanical properties of Fe0.6P scaffolds, oxygen content and carbon content (Reprinted with permission from Ref. [46]. Copyright @ 2020, Nature).

Density (g/cm³)	Compression Strength (MPa)	Young’s Modulus (GPa)
1.0	13.1 ± 1.2	0.8 ± 0.2
1.4	22.8 ± 1.1	1.3 ± 0.2

**Table 6 jfb-13-00072-t006:** Mechanical properties and corrosion behavior of Fe-Mn-(1–3 wt%)Ag alloys content (Reprinted with permission from Ref. [10]. Copyright @ 2018, Elsevier).

Samples	Density (g/cm^3^)	Ultimate Shear Stress (MPa)	Micro-Hardness (HV)	E_corr_ (mV)	*i_corr_* (µA/cm^2^)	Corrosion Rate (mm/year)
Fe-30Mn	5.49	172 ± 7	119 ± 8	−213	800	2.61
Fe-30Mn-1Ag	6.2	360 ± 5	156 ± 10	−303	860	2.49
Fe-30Mn-3Ag	6.92	490 ± 10	174 ± 10	−371	890	2.31

**Table 7 jfb-13-00072-t007:** Mechanical characterization and electrochemical parameters of iron and iron matrix with Ag or Au (Reprinted with permission from Ref. [12]. Copyright @ 2015, Wiley Online Library).

Materials	Density (g/cm³)	Average Grain Size (µm)	Compressive Yield Strength (MPa)	Ecorr (V)	Icorr (µA/cm²)	Vcorr (mm/a)
As-cast pure iron	7.831	140	~120	−0.7272	3.7416	0.0435
As-sintered pure iron	7.746	17	~250	−0.8596	6.0179	0.0709
Fe-2Ag	7.807	16	~210	−0.8412	10.188	0.1196
Fe-5Ag	7.870	17	~360	−0.8558	12.166	0.1403
Fe-10Ag	7.945	17	~200	−0.8909	15.189	0.1746
Fe-2Au	7.841	7.5	~350	−0.8095	14.967	0.1736
Fe-5Au	7.984	12	~240	−0.7959	11.498	0.1309
Fe-10Au	8.224	13	~350	−0.7791	8.833	0.0981

**Table 8 jfb-13-00072-t008:** Iron alloys with manganese and copper and electrochemical results (Reprinted with permission from Ref. [11]. Copyright @ 2019, Elsevier).

Fe-Alloy	I_corr_ (µA/cm^2^)	V_corr_ (mV)	Corrosion Rate (mmpy)
Fe-35Mn-0Cu	3.66	−678	0.043
Fe-34Mn-1Cu	2.69	−715	0.032
Fe-32Mn-3Cu	2.02	−718	0.024
Fe-30Mn-5Cu	2.88	−715	0.036
Fe-25Mn-10Cu	20.00	−600	0.258

**Table 9 jfb-13-00072-t009:** Body fluids composition, by ASTM F2129-17b (Adapted/Reprinted with permission from Ref. [51]. Copyright @ 2019, MDPI).

Component	Serum (mg/L)	Synovial Fluid (mg/L)	Interstitial Fluid (mg/L)
Cl^−^	3581	3811	4042
Na^+^	3265	3127	3280
HCO_3_^−^	1648	1880	1892
Organic acids	210	-	245
K^+^	156	156	156
Ca^2+^	100	60	100
HPO_4_^2−^	96	96	96
SO_4_^2−^	48	48	48
Mg^2+^	24	-	24
Protein	66,300	15,000	4144

**Table 10 jfb-13-00072-t010:** Simulated body fluids (SBFs) composition, by ASTM F2129-17b 17b (Adapted/Reprinted with permission from Ref. [51]. Copyright @ 2019, MDPI).

Component	Hank’s (g/L)	PBS (g/L)	Ringer’s (g/L)
NaCl	8.00	8.00	8.60
KCl	0.40	0.20	0.30
NaHCO_3_	0.35	-	-
CaCl_2_	0.14	-	-
Na_2_HPO_4_ 12H_2_O	0.12	-	-
NaH_2_PO_4_	-	1.15	-
KH_2_PO_4_	0.06	0.20	-
MgCl_2_ 6H_2_O	0.10	-	-
MgSO_4_ 7H_2_O	0.10	-	-
Phenol red	0.02	-	-
Glucose	1.00	-	-

**Table 11 jfb-13-00072-t011:** Electrochemical corrosion parameters for Pt discs in iron matrix and pure iron. (Reprinted with permission from Ref. [48]. Copyright @ 2016, Elsevier).

Materials	Corrosion Rate (mm/year)	Icorr (µA/cm²)	Ecorr (V)	Corrosion Rate (mg/cm²day)	
				Electrochemical Test	Immersion Test for 42 Days
Pt discs (Φ 4 µm × S4 µm)	0.22256	19.754	−0.88616	0.47927	0.38324
Pt discs (Φ 20 µm × S5 µm)	0.20565	17.698	−0.76282	0.44285	0.34565
Pure iron, no coating	0.11204	9.642	−0.69932	0.24127	0.14853

**Table 12 jfb-13-00072-t012:** Corrosion current density and corrosion potential of different Fe-alloys, calculated by Tafel analysis (Reprinted with permission from Ref. [32]. Copyright @ 2016, Elsevier).

Alloying Elements	I_corr_ [µA/cm^2^]	E_corr_ [V]
Fe-Mn	1.00 ± 0.06	−0.72 ± 0.04
Fe-Mn-1Ca	2.12 ± 0.92	−0.71 ±0.02
Fe-Mn-2Ca	6.36 ± 1.75	−0.66 ± 0.02
Fe-Mn-1Mg	5.89 ± 0.80	−0.65 ± 0.02
Fe-Mn0-2Mg	9.16 ± 1.25	−0.64 ±0.03

## Data Availability

Not applicable.

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
