# Peer review of "Biodegradable Iron and Porous Iron: Mechanical Properties, Degradation Behaviour, Manufacturing Routes and Biomedical Applications"

_jfb, 2022, doi:10.3390/jfb13020072_

Round 1
Reviewer 1 Report
The present review focuses on biodegradable iron and porous iron alloys, covering the areas of manufacturing to bio-applications. The authors connect nicely the mechanical properties obtained from various manufacturing techniques with the biodegradation in physiological conditions and cytocompatibility of iron-based alloys. The selected references are recent and appropriate, adequately covering the review topic.
Even though there are recent reviews dealing the corrosion and biocompatibility aspects of biodegradable porous iron alloys ( Md Yusop et al https://doi.org/10.1002/jbm.b.34893 and Materials 2021, 14(12), 3381; https://doi.org/10.3390/ma14123381) this review has a special focus on the production of iron and porous iron with advanced manufacturing techniques, making it highly interesting for the relevant scientific community.
Prior publication an extensive proofreading should be performed throughout the manuscript as some references are missing (lines 528, 554 and 571), figures 11 and 12 are not mentioned in the text and in lines 655-7 the legend of figure 13 appears, leaving the sentence incomplete and making a bit hard to follow in that section.
Moreover, the advantages of additive manufacturing techniques should be emphasized more in the conclusions section, making more clear the scope of the review.
Author Response
Response to Referees Letter (Manuscript Ref. Nº: jfb-1709067).
Dear Editor
Thanks very much for the the reviewer’s comments concerning our manuscript entitled “Biodegradable Iron and Porous Iron: mechanical properties, degradation behaviour, manufacturing routes and biomedical applications” (Manuscript Ref. Nº: jfb-1709067).
We sincerely thank the reviewer comments, which gives us a chance to improve our study and manuscript. Those comments are valuable and very helpful for revising and improving our paper. We have studied comments carefully and have made correction, which we hope to meet with approval.
The text was changed to provide the answers to the questions and comments. All changes in the text are marked in red.
Reviewer: 1
The authors thank the reviewer for the comments and suggestions that help to improve the paper.
Author's Reply to the Review Report (Reviewer 1)
Comments and Suggestions for Authors
The present review focuses on biodegradable iron and porous iron alloys, covering the areas of manufacturing to bio-applications. The authors connect nicely the mechanical properties obtained from various manufacturing techniques with the biodegradation in physiological conditions and cytocompatibility of iron-based alloys. The selected references are recent and appropriate, adequately covering the review topic.
Even though there are recent reviews dealing the corrosion and biocompatibility aspects of biodegradable porous iron alloys ( Md Yusop et al https://doi.org/10.1002/jbm.b.34893 and Materials 2021, 14(12), 3381; https://doi.org/10.3390/ma14123381) this review has a special focus on the production of iron and porous iron with advanced manufacturing techniques, making it highly interesting for the relevant scientific community.
Response: The two references were introduced, as well as other references published in 2022.
Prior publication an extensive proofreading should be performed throughout the manuscript as some references are missing (lines 528, 554 and 571), figures 11 and 12 are not mentioned in the text and in lines 655-7 the legend of figure 13 appears, leaving the sentence incomplete and making a bit hard to follow in that section.
Response: It was corrected. References were introduced and figures are now mentioned in the text.
Moreover, the advantages of additive manufacturing techniques should be emphasized more in the conclusions section, making more clear the scope of the review.
Response: The title was changed in accordance and conclusions were re-written, being manufacturing techniques emphasized.

Reviewer 2 Report
The review manuscript provides a good overview over iron and porous iron as biodegradable metal and covers an important topic. The review is based on mechanical properties, degradation behaviour and biomedical applications. The paper is well written and structured. There are two major and two minor concerns – here adjusting and improvement is necessary before publishing.
[1 - major] The title “Biodegradable Iron and Porous Iron: mechanical properties, degradation behaviour and biomedical applications” does not give the impression that the paper is strongly based to additive manufacturing processes with tailored geometries. I highly recommend to add the part of the methods to the title! In your abstract you mention that you comparison of iron produced by traditional methods.
“Biodegradable Iron and Porous Iron: mechanical properties, degradation behavior, manufacturing routes and biomedical applications”
[2 - minor]
The abstract should be extended and not be separated by an empty line (see line 10). There the structure of your review paper should be reflected in your abstract.
[3 - minor] The introduction should provide a little description of properties of Mg and Zn in comparison to Fe.
[4 - major]
You mentioned yourself the number of manuscripts on the basis of the keywords “biodegradable metals” “biodegradable porous iron” – also for 2021 and already for 2022. However, you are not reviewing any paper from 2021 and 2022. Here you need to extend you literature review!
Author Response
Response to Referees Letter (Manuscript Ref. Nº: jfb-1709067).
Dear Editor
Thanks very much for the the reviewer’s comments concerning our manuscript entitled “Biodegradable Iron and Porous Iron: mechanical properties, degradation behaviour, manufacturing routes and biomedical applications” (Manuscript Ref. Nº: jfb-1709067).
We sincerely thank the reviewer comments, which gives us a chance to improve our study and manuscript. Those comments are valuable and very helpful for revising and improving our paper. We have studied comments carefully and have made correction, which we hope to meet with approval.
The text was changed to provide the answers to the questions and comments. All changes in the text are marked in red.
Reviewer: 2
The authors thank the reviewer for the comments and suggestions that help to improve the paper.
Author's Reply to the Review Report (Reviewer 2)
Comments and Suggestions for Authors
The review manuscript provides a good overview over iron and porous iron as biodegradable metal and covers an important topic. The review is based on mechanical properties, degradation behaviour and biomedical applications. The paper is well written and structured. There are two major and two minor concerns – here adjusting and improvement is necessary before publishing.
[1 - major] The title “Biodegradable Iron and Porous Iron: mechanical properties, degradation behaviour and biomedical applications” does not give the impression that the paper is strongly based to additive manufacturing processes with tailored geometries. I highly recommend to add the part of the methods to the title! In your abstract you mention that you comparison of iron produced by traditional methods.
“Biodegradable Iron and Porous Iron: mechanical properties, degradation behavior, manufacturing routes and biomedical applications”
Response: The title was changed as suggested by the Reviewer.
[2 - minor]
The abstract should be extended and not be separated by an empty line (see line 10). There the structure of your review paper should be reflected in your abstract.
Response: The empty line was deleted and the abstract was expanded considering the structure of the review.
[3 - minor] The introduction should provide a little description of properties of Mg and Zn in comparison to Fe.
Response: A new text was added with the comparison between the properties of Mg, Zn and Fe.
[4 - major]
You mentioned yourself the number of manuscripts on the basis of the keywords “biodegradable metals” “biodegradable porous iron” – also for 2021 and already for 2022. However, you are not reviewing any paper from 2021 and 2022. Here you need to extend you literature review!
Response: The reviewer is correct. We added new references, such as Yusop et al. 2021, Gasio et al. 2021,Dong et al.2021, Roman et al. 2022, Gorejova et al 2022.

Reviewer 3 Report
- This review article contains widely collected information on biodegradable iron base alloys, it is worthwhile to compile and publish. However, which is rather biased to porous materials and cardiovascular applications. Mechanical properties and biodegradation are mainly cited from the studies on porous iron and iron base alloys. On the other hand, the author has mentioned also orthopedic applications such as fracture fixation device. Although the biodegradable metals are one of ideal implant materials, the required material properties are different between soft tissue repair in cardiovascular system and hard tissue repair in skeletal system. Even if the porous metal material has high compression strength, and the low elastic modulus close to the human bone, which cannot be applied to the fracture fixation device because the fracture repair requires the stable fixation with sufficient bending / tensile strength and fatigue strength for a certain time period. In this point of view, the review contents should be clearly separated into the cardiovascular application and the orthopedic application, alternatively focused on the cardiovascular application.
- Because of so many hyphenated words, it is difficult to smoothly read this article. It is recommended to reorganize the paragraphs by word wrap and justification to minimize the hyphenations. Also, this review article is lengthy. It is recommended to reduce the text volume by means of omitting some paragraphs.
- Line 263 and line 274 of Figure 4, the author described ‘before and after degradation’, what is this degradation? The author should clearly describe this degradation. (ref.[27])
- Please check and unify the notation of SUS316L Stainless Steel.
- Line 656 316L stainless steel
- Line 659 316SS
- Line 692 austenitic SS
- Line 722 316L SS
- Figure numbers are requested in the main text.
- Line 528 Error! Reference source not found. (Figure 11)
- Line 571Error! Reference source not found (Figure 12)
- Line 750 Figure 14
- Line 779 Figure 15
- Line 787 Figure 16
- Line 849 Figure 17
- Line 865 Figure 18
- Line 889 Figure 19
- Line 901 Figure 20
- It seems that the conclusions are abstractive and not derived from the contents of this review article. The author should describe not only the promising aspects, but also the unsolved problems and shortcomings of the reported biodegradable iron materials. Then the author concludes with further developments and prospective applications.
Author Response
Response to Referees Letter (Manuscript Ref. Nº: jfb-1709067).
Dear Editor
Thanks very much for the the reviewer’s comments concerning our manuscript entitled “Biodegradable Iron and Porous Iron: mechanical properties, degradation behaviour, manufacturing routes and biomedical applications” (Manuscript Ref. Nº: jfb-1709067).
We sincerely thank the reviewer comments, which gives us a chance to improve our study and manuscript. Those comments are valuable and very helpful for revising and improving our paper. We have studied comments carefully and have made correction, which we hope to meet with approval.
The text was changed to provide the answers to the questions and comments. All changes in the text are marked in red.
Reviewer: 3
The authors thank the reviewer for the comments and suggestions that help to improve the paper.
Author's Reply to the Review Report (Reviewer 3)
Comments and Suggestions for Authors
- This review article contains widely collected information on biodegradable iron base alloys, it is worthwhile to compile and publish. However, which is rather biased to porous materials and cardiovascular applications. Mechanical properties and biodegradation are mainly cited from the studies on porous iron and iron base alloys. On the other hand, the author has mentioned also orthopedic applications such as fracture fixation device. Although the biodegradable metals are one of ideal implant materials, the required material properties are different between soft tissue repair in cardiovascular system and hard tissue repair in skeletal system. Even if the porous metal material has high compression strength, and the low elastic modulus close to the human bone, which cannot be applied to the fracture fixation device because the fracture repair requires the stable fixation with sufficient bending / tensile strength and fatigue strength for a certain time period. In this point of view, the review contents should be clearly separated into the cardiovascular application and the orthopedic application, alternatively focused on the cardiovascular application.
Response: The reviewer is correct. It was confusing. We have now clarified the text. This type of implants can only be applied in non-load bearing orthopedic applications, which excludes locations where high strengths are required.
- Because of so many hyphenated words, it is difficult to smoothly read this article. It is recommended to reorganize the paragraphs by word wrap and justification to minimize the hyphenations. Also, this review article is lengthy. It is recommended to reduce the text volume by means of omitting some paragraphs.
Response: It was corrected.
- Line 263 and line 274 of Figure 4, the author described ‘before and after degradation’, what is this degradation? The author should clearly describe this degradation. (ref.[27])
Response: It means degradation upon immersion in simulated body fluids. A new text was also added to clarify this at the introduction: “Iron degradation occurs by a corrosion mechanism, due to electrochemical dissolution, which occurs when a metallic sample is in contact with the human body fluids.”
- Please check and unify the notation of SUS316L Stainless Steel.
- Line 656 316L stainless steel
- Line 659 316SS
- Line 692 austenitic SS
- Line 722 316L SS
Response: It was corrected in all manuscript.
- Figure numbers are requested in the main text.
- Line 528 Error! Reference source not found. (Figure 11)
- Line 571Error! Reference source not found (Figure 12)
- Line 750 Figure 14
- Line 779 Figure 15
- Line 787 Figure 16
- Line 849 Figure 17
- Line 865 Figure 18
- Line 889 Figure 19
- Line 901 Figure 20
Response: It was corrected. All missing references text figures numbers has been added to the manuscript.
- It seems that the conclusions are abstractive and not derived from the contents of this review article. The author should describe not only the promising aspects, but also the unsolved problems and shortcomings of the reported biodegradable iron materials. Then the author concludes with further developments and prospective applications.
Response: Conclusions and future trends were re-written in accordance to the Reviewer suggestions.

Reviewer 4 Report
Paper : Biodegradable Iron and Porous Iron: mechanical properties, degradation behaviour and biomedical applications present many interesting results in the field of metallic materials for medical applications as a review of literature. The authors can insert also a short comparison with the corrosion properties of as-cast Fe biodegradable materials (for example given in article https://doi.org/10.3390/met8070541, ) and highlight the advantages of porous materials, 3D printed etc. Few small corrections must be done in order to improve the review article quality:
L8: a better structure of the abstract is necessary to highlight the article findings (mention the general structure of the paper, like Fabrication, Properties, etc.)
L229: in table 2 set the information in a better table place: Young’s modulus – 200 GPa, density – 7.8 g/cm3 - with upper script, and double check the value of corrosion rate 25 mm/year
L528: Check error ! - probably figure 11
L534: mention Figure 11 in text, explain what represent the yellow stains
L571: check error message: Error! Reference source not found.
L576:better Pourbaix diagrams quality is necessary
L750: Figure 14
L766: English language check
L783: in caption of Figure 15: and weight loss
L849:Figure 17
L865:Figure 18
L901: Figure 20 - check the figures mention in text from figure 14 up to 20
L909: better structure the conclusions and literature future trends
Author Response
Response to Referees Letter (Manuscript Ref. Nº: jfb-1709067).
Dear Editor
Thanks very much for the the reviewer’s comments concerning our manuscript entitled “Biodegradable Iron and Porous Iron: mechanical properties, degradation behaviour, manufacturing routes and biomedical applications” (Manuscript Ref. Nº: jfb-1709067).
We sincerely thank the reviewer comments, which gives us a chance to improve our study and manuscript. Those comments are valuable and very helpful for revising and improving our paper. We have studied comments carefully and have made correction, which we hope to meet with approval.
The text was changed to provide the answers to the questions and comments. All changes in the text are marked in red.
Reviewer: 4
The authors thank the reviewer for the comments and suggestions that help to improve the paper.
Author's Reply to the Review Report (Reviewer 4)
Comments and Suggestions for Authors
Paper : Biodegradable Iron and Porous Iron: mechanical properties, degradation behaviour and biomedical applications present many interesting results in the field of metallic materials for medical applications as a review of literature. The authors can insert also a short comparison with the corrosion properties of as-cast Fe biodegradable materials (for example given in article https://doi.org/10.3390/met8070541, ) and highlight the advantages of porous materials, 3D printed etc.
Response: The paper by Cimpoesu et al 2018 was cited and a new text was added.
Few small corrections must be done in order to improve the review article quality:
L8: a better structure of the abstract is necessary to highlight the article findings (mention the general structure of the paper, like Fabrication, Properties, etc.)
Response: The title was changed and the abstract was re-written.
L229: in table 2 set the information in a better table place: Young’s modulus – 200 GPa, density – 7.8 g/cm3 - with upper script, and double check the value of corrosion rate 25 mm/year
Response: It was corrected.
L528: Check error ! - probably figure 11-
Response: It was corrected.
L534: mention Figure 11 in text, explain what represent the yellow stains
Response: A new text was added “The reactions lead to formation of a protective metal oxide layer on the surface (yellow spots). The interactions with the body fluids may lead to deposition of calcium phosphate on the metal oxide layer, which permit that cells adhere on the surface to form tissues [58].”
L571: check error message: Error! Reference source not found.
Response: It was corrected.
L576:better Pourbaix diagrams quality is necessary
Response: The Pourbaix diagrams, were updated with improved quality
L750: Figure 14
Response: It is now mentioned in the text.
L766: English language check
Response: A grammatical check was made on the manuscript and English was checked.
L783: in caption of Figure 15: and weight loss
Response: It was corrected.
L849:Figure 17
Response: It was corrected.
L865:Figure 18
Response: It was corrected.
L901: Figure 20 - check the figures mention in text from figure 14 up to 20
Response: It was corrected.
L909: better structure the conclusions and literature future trends – The conclusions and future trends were re-written.
